# Human CD29+/CD56+ myogenic progenitors display tenogenic differentiation potential and facilitate tendon regeneration

Xiexiang Shao[1†], Xingzuan Lin[2†], Hao Zhou[1†], Minghui Wang[3†], Lili Han[4], Xin Fu[1], Sheng Li[1], Siyuan Zhu[1], Shenao Zhou[4], Wenjun Yang[1*], Jianhua Wang[1*], Zhanghua Li[5*], Ping Hu[1,4,6,7*]

[1]Department of Orthopedic Surgery, Xinhua Hospital Affiliated to Shanghai Jiao Tong University School of Medicine, Shanghai, China; [2]Department of Sports Medicine, Peking University Third Hospital, Institute of Sports Medicine of Peking University, Beijing Key Laboratory of Sports Injuries, Beijing, China; [3]Department of Bioinformatics, Fujian Key Laboratory of Medical Bioinformatics. School of Medical Technology and Engineering. Fujian Medical University, Fuzhou, China; [4]State Key Laboratory of Cell Biology, Shanghai Institute of Biochemistry and Cell Biology, Center for Excellence in Molecular Cell Science, Chinese Academy of Sciences, Shanghai, China; [5]Department of Orthopedic Surgery, Wuhan Third Hospital, Tongren Hospital of Wuhan University, Wuhan, China; [6]Guangzhou Laboratory, Guangzhou, China; [7]Institute for Stem Cell and Regeneration, Chinese Academy of Sciences, Beijing, China

*For correspondence:
wjyang@sibcb.ac.cn (WY);
wangjianhua@xinhuamed.com.cn (JW);
lizhanghua_123@163.com (ZL);
hup@sibcb.ac.cn (PH)

†These authors contributed equally to this work

## eLife Assessment

The authors demonstrate the **valuable** discovery that human CD29+/CD56+ myogenic progenitors can differentiate into tendon through the TGFβ pathway, addressing mouse and human interspecies differences in regard to the potential of muscle stem cells. The in vivo transplantation experiments provide **convincing** evidence for the conclusion as human CD29+/CD56+ myogenic progenitors contribute to tendon regeneration, resulting in functional recovery in mouse model. The authors' approach can be used for the development of cell therapy for tendon-injured patients.

**Abstract** Tendon injury occurs at high frequency and is difficult to repair. Identification of human stem cells being able to regenerate tendon will greatly facilitate the development of regenerative medicine for tendon injury. Genetic and functional analyses identify human CD29+/CD56+ myogenic progenitors with tenogenic differentiation potential in vitro and in vivo. Transplantation of human CD29+/CD56+ myogenic progenitors contributes to injured tendon repair and thus improves locomotor function. Interestingly, the tendon differentiation potential in mouse muscle stem cells is minimal and the higher TGFβ signaling level may be the key for the distinct feature of human CD29+/CD56+ myogenic progenitors. The discovery of bi-potential CD29+/CD56+ myogenic progenitors highlights their potential as a novel adult stem cell source for tendon regeneration.

## Introduction

Skeletal muscle is a tissue with great regeneration ability due to the existence of muscle stem cells (MuSCs). MuSCs are adult stem cells located at the periphery of myofibers between the basal lamina and the plasmalemma of the myofibers and responsible for muscle regeneration (*Fu et al., 2015a*; *Relaix et al., 2021*). MuSCs have been considered to be unipotent stem cells to have a sole differentiation potential to myofibers (*Seale et al., 2001*). MuSCs undergo expansion and differentiate to multinuclei myofibers after injury in vivo. MuSCs have remarkable abilities to support muscle regeneration. After transplantation, the engrafted mouse MuSCs go through active expansion and regenerate myofibers (*Collins et al., 2005*). Isolation by FACS and expansion of MuSCs have been reported from several species, including mouse, pig, and human (*Ding et al., 2017*; *Shao et al., 2023*). Single-cell sequencing analysis from human skeletal muscles has also revealed the existence of Pax7+ myogenic progenitors (*Shao et al., 2023*; *Barruet et al., 2020*). Due to the different motion patterns, the regeneration capacity may be different between humans and rodents. Current investigations have suggested that human myogenic progenitor cells did not share same markers as that in mice (*Boldrin et al., 2010*), and the expression pattern of oxidative enzymes and cytokines between these two species is also different (*Bareja et al., 2014*), suggesting that human muscle progenitor cells may have distinct features from mice.

FACS method to isolate cells with myogenic differentiation potential from human muscle biopsies has been established. It has been reported that CD56 (NCAM)+ and CD56+CD29+ cells from skeletal muscle display myogenic differentiation potential (*Bareja et al., 2014*; *Spinazzola and Gussoni, 2017*; *Pisani et al., 2010b*; *Xu et al., 2015*). The CD56+CD34+ progenitor cells isolated from human skeletal muscle biopsies have been reported to have chondrogenic, adipogenic, and osteogenic potentials besides myogenic potential (*Zheng et al., 2007*; *Castiglioni et al., 2014*; *Pisani et al., 2010a*). CD56+CD34- progenitor cells are free of adipogenic potential (*Pisani et al., 2010b*). The differentiation potential of human myogenic progenitors remains to be further explored.

Skeletal muscle directly connects to tendons, which is responsible for transmitting forces from skeletal muscle to bone to generate active movement. Tendinopathy affects more than 10% of the population under 45 and compromises the tendon functions (*Gaida et al., 2010*; *Voleti et al., 2012*). Tendon injury healing is slow and incomplete due to the low number of cells in the tendon and the hypovascular and anaerobic environment (*Korcari et al., 2023*; *Pennisi, 2002*). Tendon stem/progenitor cells (TDSCs) is a cell population derived from tendon and considered to be a subgroup of mesenchymal stem cells that have abilities to improve tendon injury healing (*Bi et al., 2007*; *Kohler et al., 2013*). However, the number of TDSCs in the tendon is low, and retrieving TDSCs is invasive and exacerbates tendon injury. Morphology and proliferation ability loss in the culture system also hampers the efforts to obtain sufficient amounts of active TDSCs. Finding more cell types supporting tendon regeneration will facilitate the development of regenerative medicine to treat tendon injury.

The activity of tenogenesis is tightly regulated by many signaling pathways, especially for TGFβ signaling (*Nourissat et al., 2015*). TGFβ signaling is indispensable for tendon development (*Pryce et al., 2009*), and it could systematically promote the tenogenic differentiation of stem cells (*Yang et al., 2017*; *Barsby and Guest, 2013*). The downstream effectors SMAD2 and SMAD3 are able to activate the transcription of tendon-specific genes, which further facilitates tendon development and tenogenic differentiation (*Nourissat et al., 2015*; *Berthet et al., 2013*; *Wang et al., 2020*). After tendon injury, TGFβ signaling promotes the proliferation, migration, and differentiation of TDSCs (*Li et al., 2021*), increases tendon collagen synthesis (*You et al., 2020*), and contributes to matrix anabolism for tendon remodeling (*Jones et al., 2013*).

Here, human CD29+/CD56+ myogenic progenitors exhibit tenogenic differentiation capacity in both in vitro and in vivo settings. Transplantation of human CD29+/CD56+ myogenic progenitors to injured tendon in mice improved the tendon regeneration, suggesting its bipotential ability. Interestingly, the tendon differentiation potential in mouse MuSCs is minimal, and the higher TGFβ signaling level may be the key for the distinct feature of human CD29+/CD56+ myogenic progenitors.

## Results

### The CD29+/CD56+ cells isolated from human muscle biopsies display robust features of myogenic progenitors

To analyze the cell components of human skeletal muscle, single-cell sequencing analysis was performed using skeletal muscle biopsy. A total of 57,193 cells were included for analysis. The cells were grouped to nine cell populations (*Figure 1a and b*). Consistent with the previous single-cell sequencing results from human muscles (*Rubenstein et al., 2020*), fibro/adipogenic progenitors (FAPs), endothelial cells, myocytes, and myogenic progenitors were among the identified cell types (*Figure 1a and b*). Especially, myogenic progenitors accounted for 12.4% of all the mononuclear cells, while tenocytes only accounted for 0.06% (*Figure 1c*). To investigate the identity of CD29+/CD56+ cells, joint expression analysis was performed (*Figure 1d*). The scRNA-seq data revealed that all the CD29+/CD56+ cells were myogenic progenitors, which occupied 19.3% of all the myogenic progenitors (*Figure 1e*). However, there existed no tenocytes with CD29+/CD56+ (*Figure 1d*). Combined, the scRNA-seq data revealed human CD29+/CD56+ cells were myogenic progenitors.

To confirm the single-cell analysis results, fluorescence-activated cell sorting (FACS) was used to isolate human CD29+/CD56+ myogenic progenitors from muscle biopsies (*Shao et al., 2023*; *Xu et al., 2015*). CD31-/CD45-/CD29+/CD56+ cells were collected from the single-cell suspension dissociated from muscle tissue. The collected cells were stained with anti-PAX7, anti-MYOD1, and anti-MYF5 antibody to confirm the purity of isolated CD29+/CD56+ cells, and nearly all of the obtained cells were positively stained with these myogenic progenitors markers (*Figure 1f and g*). The proliferation assay indicated high in vitro expansion ability of these human CD29+/CD56+ myogenic progenitors (*Figure 1h*). When induced to differentiate by 0.4% Ultroser G, human CD29+/CD56+ myogenic progenitors differentiate to myotubes robustly. Approximately 90% of nuclei were present in myosin heavy chain (MyHC)-positive myotubes (*Figure 1i and j*). Consistently, the expression of *PAX7*, *MYF5*, and *MYOD1* was enriched in the myogenic progenitors, and their expression decreased after differentiation as shown by RT-qPCR assays (*Figure 1k*). In contrast, the expression of genes marking differentiation such as myogenin (*MYOG*), myosin heavy chain 1 (*MYH1*), and myosin heavy chain 3 (*MYH3*) was significantly upregulated in differentiated myotubes (*Figure 1l*). Taken together, these results suggest that the CD29+/CD56+ cells isolated from human muscle biopsies have robust features of myogenic progenitors.

### Human CD29+/CD56+ myogenic progenitors display tenogenic differentiation potential in vitro

Since previous studies indicated that there could be tenogenic differentiation ability for muscle-derived cells (*Shao et al., 2020*; *Sassoon et al., 2012*; *Chen et al., 2012*), the tenogenic capacity of CD29+/CD56+ myogenic progenitors was subsequently investigated. The isolated primary human CD29+/CD56+ myogenic progenitors were induced to differentiate to tenocytes by 100 ng/ml GDF5, 100 ng/ml GDF7, and 0.2 mM ascorbic acid for 12 days. After 12 days of tendon differentiation, the morphology of cells displayed dramatic differences from those undergoing myogenic differentiation (*Figure 2—figure supplement 1a*). Furthermore, the expression of tendon markers such as TNC and SCX was significantly increased (*Figure 2a–c*). Moreover, some other tendon-related genes, such as *COL I*, *MKX*, *THBS4*, and *COMP*, were also enriched after tendon differentiation (*Figure 2c*). In contrast, the expression of these genes was not enriched upon myogenic differentiation (*Figure 2a–c*). The expression of genes marking myogenic differentiation such as MYOG and MyHC was only detected in a small portion of cells after tenogenic differentiation (*Figure 2—figure supplement 1b and c*). Compared to the over 90% of differentiation efficiency upon myogenic differentiation, only about 20% of cells showed MYOG and MyHC expression after 12 days of tenogenic induction (*Figure 2—figure supplement 1b and c*). The genes marking differentiated myotubes such as *MYH3*, *DESMIN*, and *MYL1* showed moderate elevation after tenogenic differentiation, while dramatic upregulation of these genes was observed after myogenic differentiation (*Figure 2—figure supplement 1d and e*). These results combined suggest that human CD29+/CD56+ myogenic progenitors are capable of tendon differentiation in vitro.

To strengthen the findings, clonal analysis was performed on CD29+/CD56+ myogenic progenitors. The freshly isolated primary human CD29+/CD56+myogenic progenitors were seeded to 96-well plate

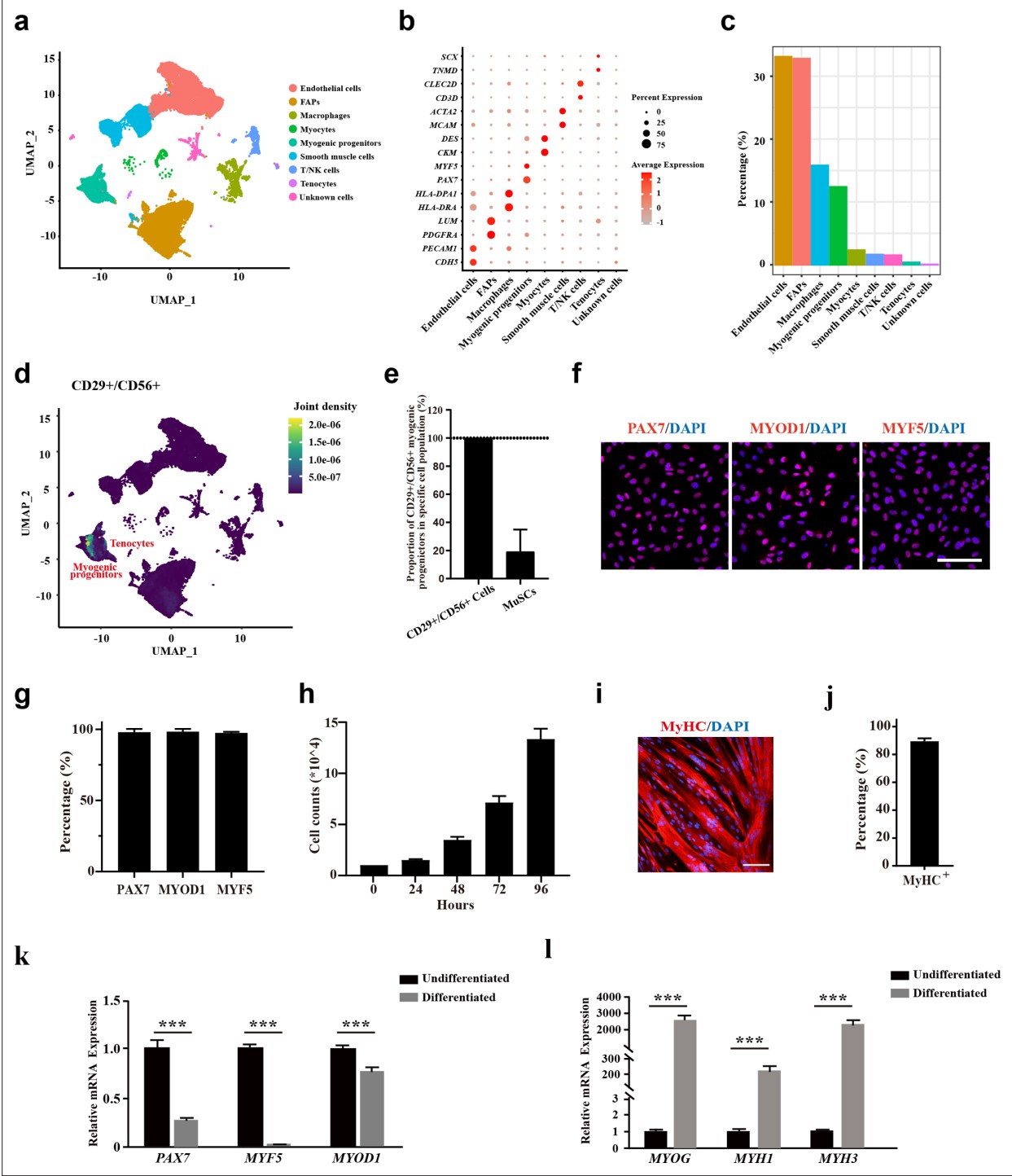

**Figure 1.** The CD29+/CD56+ cells isolated from human muscle biopsies display robust features of myogenic progenitors. (**a**) UMAP plot of all mononuclear cells isolated from human skeletal muscle which were colored by cell clusters. Three samples with a total of 57,193 cells were included for analysis. (**b**) Dot blot of representative genes in each cell cluster. (**c**) Bar plot of cell proportion for each cell cluster. (**d**) Plot of joint expression analysis for CD29+/CD56+ cells in total mononuclear cells isolated from human skeletal muscle. (**e**) Proportion of CD29+/CD56+ myogenic progenitors in specific cell population. Error bars indicate standard deviation (n=3). (**f**) Immunofluorescence staining of PAX7, MYOD1, and MYF5 in primary human CD29+/CD56+ myogenic progenitors. Scale bars, 100 μm.(**g**) Statistical analysis of the percentage of PAX7+, MYOD1+, and MYF5+ cells in the isolated CD29+/CD56+ myogenic progenitors. Error bars indicate standard deviation (n=5). (**h**) Analysis of cell proliferation for human CD29+/CD56+ myogenic progenitors in vitro. 10,000 isolated human CD29+/CD56+ myogenic progenitors were plated for proliferation and counted at each time point. Error bars indicate standard deviation (n=3). (**l**) Immunofluorescence staining of MyHC in myotubes differentiated from human CD29+/CD56+

*Figure 1 continued on next page*

*Figure 1 continued*

myogenic progenitors. Primary human CD29+/CD56+ myogenic progenitors were isolated and differentiated to myotubes for 5 days followed by MyHC immunofluorescence staining. Scale bars, 100 μm. (**j**) Statistical analysis of the percentage of nuclei in MyHC+ myotubes after differentiation of the human CD29+/CD56+ myogenic progenitors. Error bars indicate standard deviation (n=5). (**k**) Relative expression level of *PAX7*, *MYF5*, and *MYOD1* in human CD29+/CD56+ myogenic progenitors (Undifferentiated) and differentiated myotubes (Differentiated). RT-qPCR assays were performed for human CD29+/CD56+ myogenic progenitors before and after myogenic differentiation. GAPDH served as a reference gene. Error bars indicate standard deviation (n=3). \*\*\*p<0.001. (**l**) Relative expression level of *MYOG*, *MYH1*, and *MYH3* in human CD29+/CD56+ myogenic progenitors (Undifferentiated) and differentiated myotubes (Differentiated). RT-qPCR assays were performed for human CD29+/CD56+ myogenic progenitors before and after myogenic differentiation. GAPDH served as a reference gene. Error bars indicate standard deviation (n=3). \*\*\*p<0.001.

with the concentration of 1 cell/well. The wells of single cell were allowed to proliferate for 10 days followed by tenogenic or myogenic induction. The plates with myogenic induction were differentiated for 4 days, and immunofluorescence staining of MyHC was performed to determine myogenic differentiation. The plates with tenogenic induction were differentiated for 12 days, and immunofluorescence staining of SCX was performed to determine tenogenic differentiation (*Figure 2d*). The number of wells showing positive MyHC staining was counted, and the myogenic differentiation efficiency was calculated (*Figure 2e and g*). MyHC+ myotubes were observed in over 95% of wells with alive cells (*Figure 2g*). Similarly, the number of wells displaying positive SCX staining was counted and the tenogenic differentiation efficiency was calculated (*Figure 2f and g*). Approximately 40% of CD29+/CD56+ myogenic progenitors displayed tenogenic differentiation ability (*Figure 2g*), suggesting that human CD29+/CD56+ myogenic progenitors have tenogenic differentiation potential. Taken together, these results suggest that human CD29+/CD56+ myogenic progenitors have dual differentiation potentials toward muscle or tendon in vitro.

## Tenocytes differentiated from human CD29+/CD56+ myogenic progenitors display a similar expression profile to primary tenocytes

RNA sequencing was then performed to further determine the lineage of the tenocytes differentiated from human CD29+/CD56+ myogenic progenitors. Human CD29+/CD56+ myogenic progenitors were induced toward myogenic or tenogenic differentiation, respectively. Primary human tenocytes were isolated using previously established protocols (*Han et al., 2018*). These cells were subjected to RNA sequencing analysis. The differentiated myotubes and tenocytes displayed distinct expression patterns (*Figure 3a*). The differentiated tendon cells share high similarity to fresh isolated tenocytes from human tendons (*Figure 3a*), suggesting that human CD29+/CD56+ myogenic progenitors are capable of tendon differentiation. Consistently, two distinct sets of genes representing myogenic markers and tenogenic markers were upregulated after myogenic induction and tenogenic induction, respectively (*Figure 3b and c*). These results suggest that human CD29+/CD56+ myogenic progenitors are capable of dual direction differentiation.

Further GO analysis also displayed the activation of two distinct sets of cell features. Upon the skeletal muscle differentiation, terms related to skeletal muscle functions such as skeletal muscle thin filament assembly, skeletal muscle contraction, muscle organ development, and sarcomere organization were enriched (*Figure 3d*), suggesting the muscular identity of the differentiated cells. In contrast, terms related to tendon formation, tendon development, and tendon cell differentiation were enriched after tenogenic differentiation (*Figure 3e*), suggesting that tendon identity is achieved in the differentiated cells. Together, these results suggest that human CD29+/CD56+ myogenic progenitors are capable of both myogenic and tenogenic differentiation in vitro.

## Murine MuSCs display poor tenogenic differentiation ability

The presence of tenogenic differentiation potential in rodent MuSCs was subsequently investigated. Mouse MuSCs were isolated by positive marker of Vcam1 as described previously (*Shao et al., 2023*) and induced for tenogenic differentiation. In sharp contrast to human CD29+/CD56+ myogenic progenitors, murine MuSCs failed to be induced to tendon cells upon the same induction condition as that for human CD29+/CD56+ myogenic progenitors, though the myogenic differentiation is as efficient as the human CD29+/CD56+ myogenic progenitors. After myogenic differentiation, 93.9% of nuclei were present in MyHC+ myotubes (*Figure 4a and b*). In sharp contrast, no Scx+ cells were observed after 12 days of induction for tenogenic differentiation (*Figure 4a*). Consistent with the

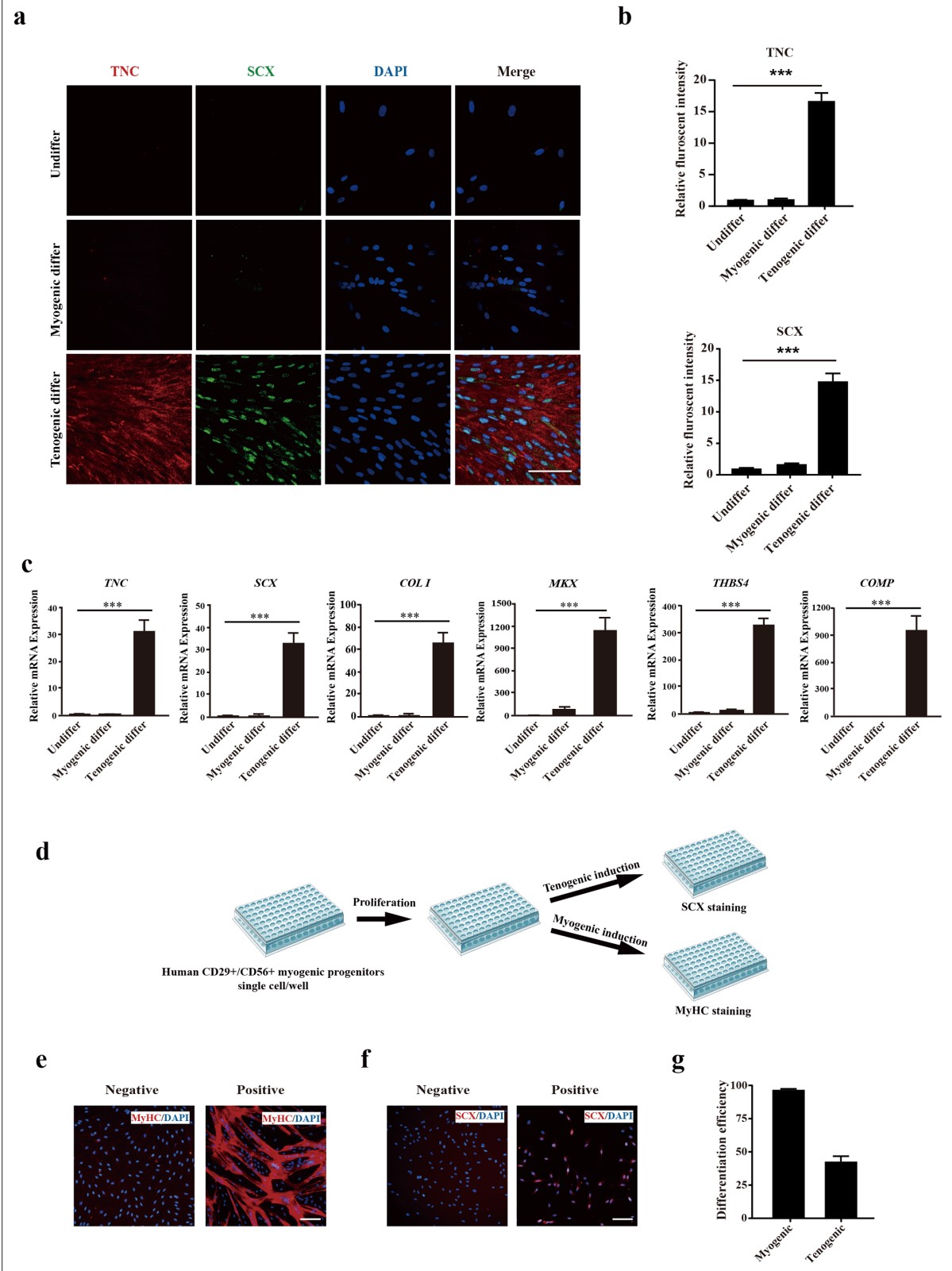

**Figure 2.** Human CD29+/CD56+ myogenic progenitors display tendon differentiation ability in vitro. (**a**) Immunofluorescence staining of tendon marker TNC and SCX in human CD29+/CD56+ myogenic progenitors induced for myogenic and tenogenic differentiation, respectively. Scale bars, 100 μm. (**b**) Quantification of TNC and SCX fluorescent intensity in human CD29+/CD56+ myogenic progenitors undergoing myogenic and tenogenic differentiation, respectively. Error bars indicate standard deviation (n=5). (**c**) Relative expression levels of genes enriched in tendon cells. RT-qPCR assays

*Figure 2 continued on next page*

*Figure 2 continued*

were performed with human CD29+/CD56+ myogenic progenitors upon myogenic and tenogenic differentiation, respectively. GAPDH served as a reference gene. Error bars indicate standard deviation (n=3). ***p<0.001. (**d**) Scheme of clonal proliferation and differentiation assay. (**e**) Representative immunofluorescence staining images of MyHC as the marker for successful myogenic differentiation. Scale bars, 100 μm. (**f**) Representative immunofluorescence staining images of SCX as the marker for successful tenogenic differentiation. Scale bars, 100 μm. (**g**) Statistical analysis of the myogenic and tenogenic differentiation efficiency of human CD29+/CD56+ myogenic progenitors. Error bars indicate standard deviation (n=3).

The online version of this article includes the following source data and figure supplement(s) for figure 2:

**Figure supplement 1.** Human CD29+/CD56+ myogenic progenitors display tenogenic differentiation potential.

**Figure supplement 1—source data 1.** Labeled raw data for TNC and MyHC of human CD29+/CD56+ myogenic progenitors which were induced towards myogenic and tenogenic differentiation.

**Figure supplement 1—source data 2.** Raw data for TNC and MyHC of human CD29+/CD56+ myogenic progenitors which were induced towards myogenic and tenogenic differentiation.

immunofluorescence staining results, RT-qPCR results revealed that myogenic differentiation marker genes such as *MyoG*, *Myh1*, and *Myh3* were significantly upregulated under both myogenic and tenogenic differentiation conditions (*Figure 4c*), suggesting that murine MuSCs predominantly commit myogenic differentiation under induction. Different from human CD29+/CD56+ myogenic progenitors, the expression of genes indicating tendon cell fate such as *Scx*, *Tnc*, *Col I*, *Mkx*, and *Thbs4* did not increase after tenogenic differentiation (*Figure 4d*), suggesting the failure to induce tenogenic cell fate from murine MuSCs. Together, these results suggest that murine MuSCs display almost no tenogenic differentiation potential in vitro.

Lineage-tracing experiments in mice were conducted to further assess the in vivo tenogenic potential of mouse MuSCs (*Figure 4e*). Pax7CreERT2 mice were crossed to flox-Stop-flox-tdTomato mice. MuSCs and the descendants of MuSCs will be labeled by tdTomato (*Figure 4—figure supplement 1a*). Tendon injury in mice was generated by mimicking the peroneus longus tendon removal surgery in humans (*Figure 4—figure supplement 1b*). In humans, it has been reported that the tendon could be regenerated to some extent after the peroneus longus tendon removal surgery based on MRI (*Shao et al., 2020*). In this surgery, injury of the skeletal muscle adjacent to the removed tendon was inevitable. The accompanied skeletal muscle injury could activate MuSCs and make them available for tendon regeneration. To further activate MuSCs to guarantee that sufficient amount of activated MuSCs were available around the tendon injury site, 15 μL of 10 μM cardiotoxin (CTX) was injected into adjacent muscle tissue to induce local muscle injury and promote MuSC activation. If MuSCs can participate tendon regeneration, tdTomato+ tendon cells would be observed after the repair of tendon injury.

Expectedly, a large amount of tdTomato+ myofibers was observed after muscle injury (*Figure 4—figure supplement 1c and d*), suggesting that the tracing system works well. Nevertheless, less than 0.2% tendon cells originated from mouse MuSCs were observed even 4 months after tendon removal (*Figure 4f–g*). These results suggest that murine MuSCs have poor tendon differentiation abilities.

## Transplantation of human CD29+/CD56+ myogenic progenitors facilitates tendon regeneration

The effect of transplanting human CD29+/CD56+ myogenic progenitors on tendon regeneration in mice was subsequently investigated. An approximately 1.5 mm long and 0.5 mm wide transverse incision was performed at 5 mm from the calcaneus in Achilles tendon for NOD/SCID immunodeficient mice. A total of 50,000 human CD29+/CD56+ myogenic progenitors packed in hydrogel were planted at the incision site (*Figure 5a*). As a control, mixture of PBS and hydrogel was transplanted in the SCID recipient mice undergoing the same tendon injury at the cleavage sites. Packing the cells with hydrogel concentrated the transplanted cells at the local injury sites. Two months after transplantation, the tendons carrying transplanted with or without human CD29+/CD56+ myogenic progenitors were harvested, respectively (*Figure 5a*).

In mice transplanted with human CD29+/CD56+ myogenic progenitors, continuous cryosections containing muscle and tendon tissues were generated. Immunofluorescence staining was performed to detect tendon and muscle markers with two cryosections adjacent to each other, respectively. The two sets of images obtained on continuous cryosections were superimposed on each other to

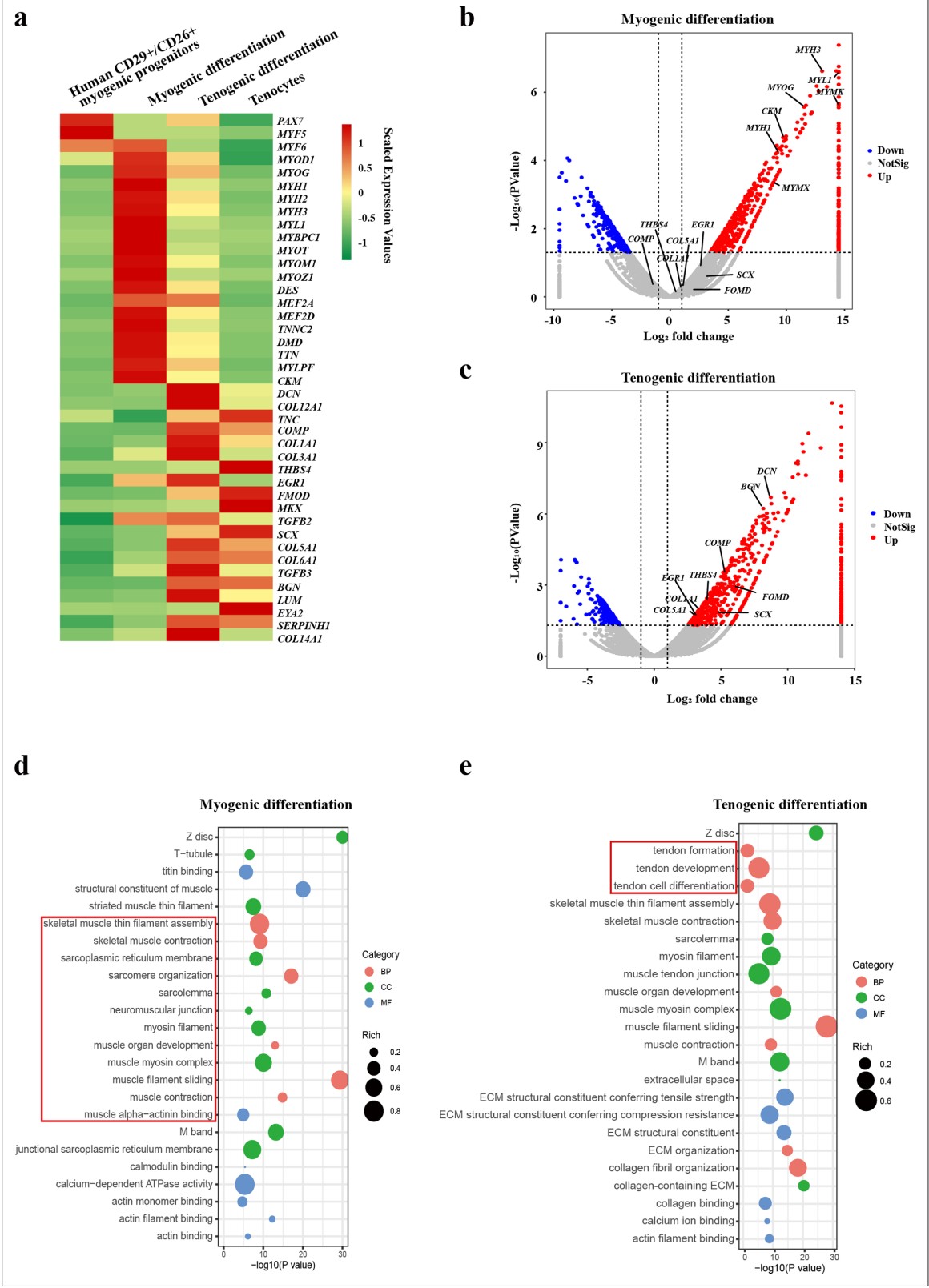

**Figure 3.** Human CD29+/CD56+ myogenic progenitors have tendon differentiation potential. (**a**) Heat map of gene expression profiles of human CD29+/CD56+ myogenic progenitors, human CD29+/CD56+ myogenic progenitors after myogenic differentiation, human CD29+/CD56+ myogenic progenitors after tenogenic differentiation, and primary tenocytes.(**b**) Volcano plot of genes enriched in myogenic differentiation of human CD29+/CD56+ myogenic progenitors. (**c**) Volcano plot of genes enriched in tenogenic differentiation of human CD29+/CD56+ myogenic progenitors.

*Figure 3 continued on next page*

*Figure 3 continued*

(**d**) Bubble chart of GO analysis of cellular process upregulated in myogenic differentiation of human CD29+/CD56+ myogenic progenitors. (**e**) Bubble chart of GO analysis of cellular process up-regulated in tenogenic differentiation of human CD29+/CD56+ myogenic progenitors.

pinpoint the position of tendons. Immunofluorescence staining of antibody specifically recognizing human TNC indicated the position of regenerated tendon-like tissue from human cells in the harvested tissues (*Figure 5b*). The presence of human cells was also illustrated by the immunofluorescence staining with the antibody specifically recognizing human Lamin A/C. In the control PBS injection group, where the mixture of PBS and hydrogel instead of human CD29+/CD56+ myogenic progenitors was injected to the incision sites, no human Lamin A/C was detected. These results confirmed the specificity of human TNC and Lamin A/C antibody. Immunofluorescence staining revealed that over 75% of the human cells showed TNC expression (*Figure 5b and d*), suggesting that the majority of the transplanted human CD29+/CD56+ myogenic progenitors differentiate to tendon cells in vivo. The immunofluorescence staining of MyHC and human Lamin A/C was performed to detect the muscle cells originated from the transplanted human CD29+/CD56+ myogenic progenitor cells. Only about 12.8% of the human cells detected expressed MyHC (*Figure 5c and d*). Moreover, the human cells were predominantly enriched at the tendon region (*Figure 5b*). Only a few MyHC+ cells originated from the transplanted human CD29+/CD56+ myogenic progenitors scattered in the muscle region (*Figure 5c*). To further confirm the tendon differentiation potential of the transplanted human CD29+/CD56+ myogenic progenitors, immunofluorescence staining of SCX and TNMD was also performed. The majority (80.0% and 74.6%) of transplanted human CD29+/CD56+ myogenic progenitors also expressed SCX and TNMD (*Figure 5—figure supplement 1a, b and c*). Furthermore, Col I was predominantly expressed in the regenerated tendon-like tissue rather than Col III after human cells transplantation (*Figure 5e*), indicating human CD29+/CD56+ myogenic progenitors could contribute to structural repair of injured tendon and facilitate the healing process. Taken together, these results suggest that human CD29+/CD56+ myogenic progenitors are capable of tendon differentiation in vivo and contributing to tendon regeneration.

In sharp contrast to human CD29+/CD56+ myogenic progenitors, when 50,000 murine MuSCs constitutively expressing tdTomato were transplanted to pre-injured tendon in NOD/SCID mice under the same condition, less than 0.3% of tdTomato+TNC+ cells were detected (*Figure 5f and g*). However, the myogenic differentiation potential of human CD29+/CD56+ myogenic progenitors and mouse MuSCs was similar. Muscle injury was induced by muscular injection of CTX in tibialis anterior (TA) muscle in NOD/SCID mice. TA muscles were irradiated to kill the local MuSCs as described previously (*Fu et al., 2015b*). Transplantation of human CD29+/CD56+ myogenic progenitors and mouse MuSCs to the irradiated pre-injured recipient mice was performed, respectively. TA muscles were harvested after 28 days. Transplantation of both human CD29+/CD56+ myogenic progenitors and murine MuSC displayed robust engraftment efficiency (*Figure 5—figure supplement 1d-i*). These results suggest that human CD29+/CD56+ myogenic progenitors and murine MuSCs have similar myogenic differentiation potential.

Combined, the above results suggest that human CD29+/CD56+ myogenic progenitors have dual differentiation potentials toward myogenesis or tenogenesis in vivo, while murine MuSCs predominantly commit myogenesis.

## Transplantation of human CD29+/CD56+ myogenic progenitors improves locomotor function after tendon injury

The impact of human CD29+/CD56+ myogenic progenitor transplantation on tendon function was subsequently evaluated. First, transmission electron microscope (TEM) analysis was used to evaluate the microstructure of the injured tendon 2 months after transplantation. Larger and denser collagen fibrils of the tendons were identified in the transplanted group than the control group with PBS injection, although the maturation level of collagen fibrils could still not reach those in the uninjured tendon (*Figure 6a and b*). The effect of human CD29+/CD56+ myogenic progenitor transplantation on tendon biomechanical properties was subsequently assessed. The max failure load and stiffness of the tendons from the transplantation group were significantly better than those from the PBS injection control group (*Figure 6c*). These results combined suggest that transplantation of human CD29+/

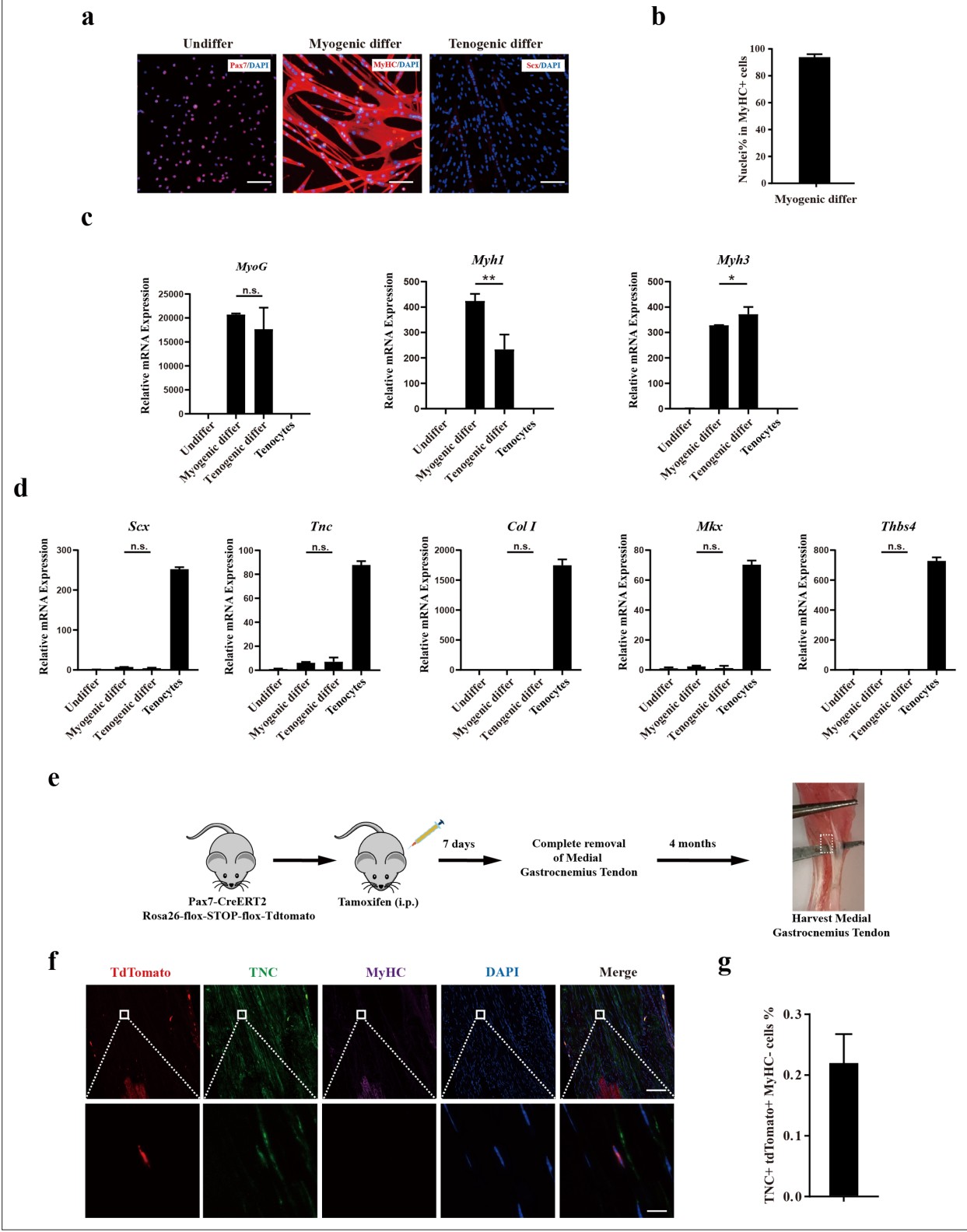

**Figure 4.** Murine muscle stem cells (MuSCs) display poor tenogenic differentiation ability. (**a**) Immunofluorescence staining of murine MuSCs directed toward myogenic and tenogenic differentiation, respectively. The undifferentiated murine MuSCs were stained with Pax7 and DAPI. Murine MuSCs after myogenic differentiation were stained with MyHC and DAPI. Murine MuSCs after tenogenic differentiation were stained with Scx and DAPI. Scale bars, 100 μm. (**b**) Statistical analysis of the percentage of nuclei in MyHC+ myotubes. Error bars indicate standard deviation (n=5). (**c**) Relative expression

*Figure 4 continued on next page*

*Figure 4 continued*

levels of myogenic differentiation-specific genes *MyoG*, *Myh1*, and *Myh3*. Mouse MuSCs were induced to differentiate toward muscle and tendon, respectively. Mouse MuSCs before and after differentiation together with primary tenocytes were harvested and subjected to RT-qPCR analysis. *Gapdh* served as a reference gene. Error bars indicate standard deviation (n=3). *p<0.05, **p<0.01, n.s. indicated s n>0.05. (**d**) Relative expression levels of tenogenic differentiation marker genes *Scx*, *Tnc*, *Col I*, *Mkx*, and *Thbs4*. Mouse MuSCs were induced to differentiate toward muscle and tendon, respectively. Mouse MuSCs before and after differentiation together with primary tenocytes were harvested and subjected to RT-qPCR analysis. *Gapdh* served as a reference gene. Error bars indicate standard deviation (n=3). n.s. indicates n>0.05. (**e**) Scheme of Pax7+ MuSC progeny lineage tracing assay. (**f**) Immunofluorescence staining of tendon tissue 4 months after injury. tdTomato+ cells indicates the progeny of Pax7+MuSCs. TNC+ cells indicate tendon cells. MyHC+ cells indicate muscle fibers. DAPI indicates nuclei staining. Merged indicates the merged images of tdTomato, TNC, MyHC, and DAPI. The upper panel indicates the low-magnification images. Scale bars, 100 μm. The lower panel indicates the high-magnification images of the region label by white square in the upper panel. Scale bars, 10 μm. (**g**) The statistical analysis of the percentage of tdTomato+TNC+ cell 4 months after tendon injury. Error bars indicate standard deviation (n=5).

The online version of this article includes the following figure supplement(s) for figure 4:

**Figure supplement 1.** Murine muscle stem cells (MuSCs) display poor tenogenic differentiation ability.

CD56+ myogenic progenitors improves the collagen fibril maturation and biomechanical property of the injured tendon.

Whether the improved tendon regeneration could facilitate the whole organism locomotor function was next investigated. Since the Achilles tendon transmits the plantarflexion force from gastrocnemius muscle to calcaneus, the plantarflexion force of involved leg was also performed 2 months after tendon injury. Expectedly, transplantation of human CD29+/CD56+ myogenic progenitors for the injured tendon also contributed to improving both twitch and tetanus plantarflexion force when compared with the PBS injection control group, although the plantarflexion force could still not reach the level of the uninjured leg (*Figure 6d*). Consistent with the improved plantarflexion force, the endurance time and max fatigue speed of the mice transplanted with human CD29+/CD56+ myogenic progenitors were better in treadmill test when compared to the PBS group (*Figure 6e*). Since immunofluorescence staining with human Lamin A/C revealed that only a small number of human CD29+/CD56+ myogenic progenitors engrafted in muscle sporadically (*Figure 5c and d*), these results suggest that the transplanted human CD29+/CD56+ myogenic progenitors can improve locomotor function by directly repairing injured tendon.

## TGFβ signaling pathway contributes to tenogenic differentiation of human CD29+/CD56+ myogenic progenitors

Since the human CD29+/CD56+ myogenic progenitors and murine MuSCs shared the same strain of recipient mice and the same tendon injury while being transplanted, they had a similar microenvironment. Therefore, the distinct differentiation potentials are due to the cell-intrinsic differences between species. Gene expression profiles of human CD29+/CD56+ myogenic progenitors were compared with those of murine MuSCs. Interestingly, TGFβ signaling was identified in KEGG enrichment analysis of upregulated genes in human CD29+/CD56+ myogenic progenitors when compared with murine MuSCs (*Figure 7a*), indicating that TGFβ signaling could be the key node for maintaining the tendon differentiation potential. Especially, SMAD2 and SMAD3 were identified in upregulated gene set which was enriched in TGFβ signaling pathway (*Figure 7b*). Since TGFβ/SMAD2/SMAD3 axis plays a crucial role in tendon development and tenogenic differentiation (*Nourissat et al., 2015*; *Pryce et al., 2009*; *Berthet et al., 2013*), these data further indicated the potential crucial role of TGFβ signaling for tenogenic differentiation ability of human CD29+/CD56+ myogenic progenitors. Then TGFβ signaling inhibitor SB-431542 was used to investigate the biological effect of TGFβ signaling during tenogenic induction. The immunofluorescent staining, western blot assay, and RT-qPCR results showed SB-431542 significantly suppressed expression level of tendon-related genes *SCX*, *TNC*, *COL I*, *MKX*, and *THBS4* (*Figure 7c–e*). On the contrary, myogenic differentiation ability of human CD29+/CD56+ myogenic progenitors was increased after treatment of SB-431542 during tenogenic induction (*Figure 7f and g*). Taken together, these data indicated that TGFβ signaling pathway contributes to tenogenic differentiation of human CD29+/CD56+ myogenic progenitors.

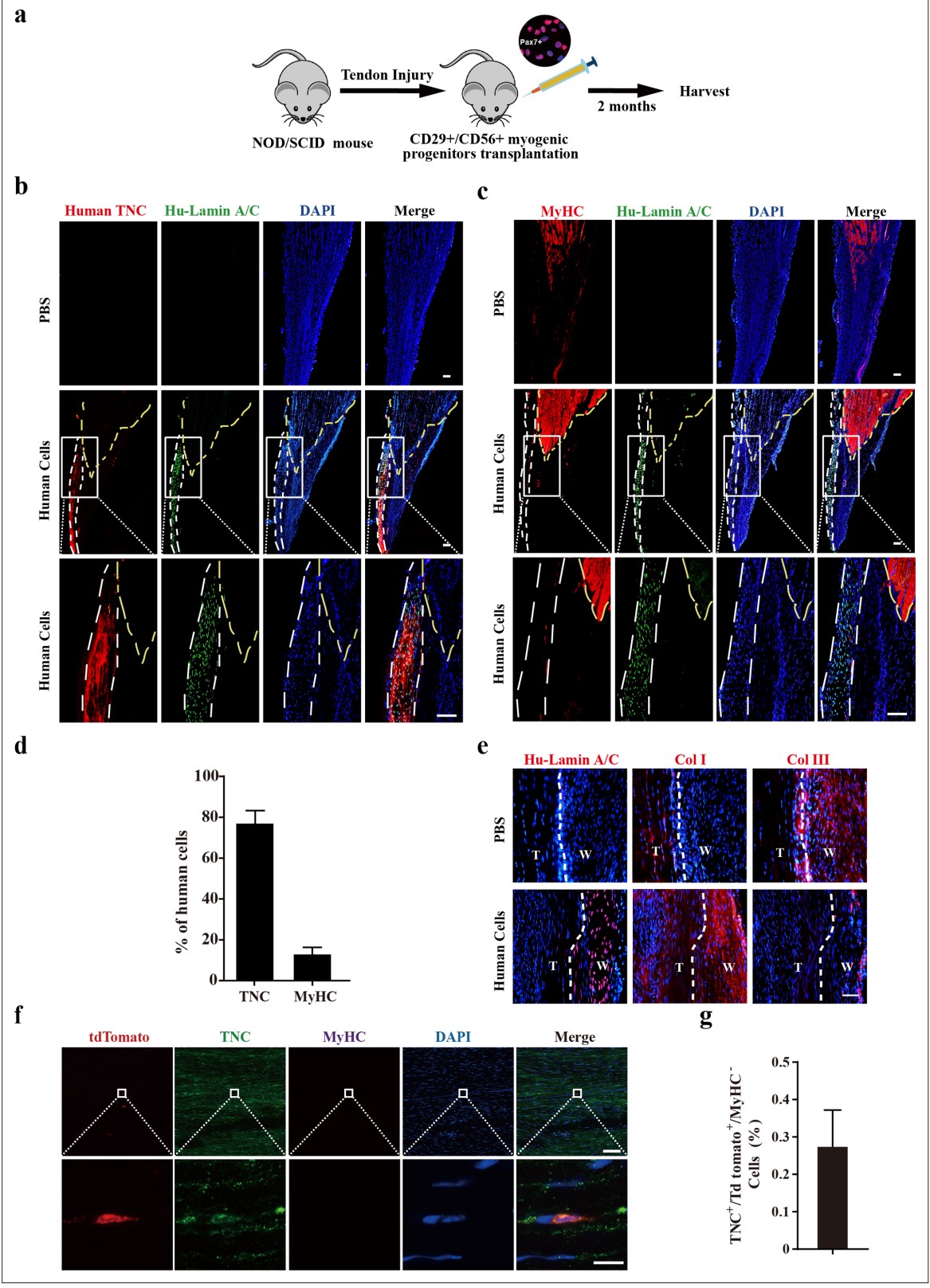

**Figure 5.** Transplantation of human CD29+/CD56+ myogenic progenitors facilitates tendon regeneration. (**a**) Scheme of transplantation of human CD29+/CD56+ myogenic progenitors for NOD/SCID with tendon injury. (**b**) Immunofluorescence staining of the regenerated tendon-like tissue after transplantation of human CD29+/CD56+ myogenic progenitors. Tendon injury was induced in recipient NOD/SCID mice, and 50,000 human CD29+/CD56+ myogenic progenitors were transplanted to the injured tendon at the injured site. The regenerated tendon-like tissue, the connected

*Figure 5 continued on next page*

*Figure 5 continued*

muscle, and the surrounding soft tissues were harvested to make continuous cryosections. One of the continuous cryosections was subjected to immunofluorescence staining of tendon marker human-specific TNC and human-specific Lamin A/C. DAPI indicates nuclei staining. Merge indicates the merged images of human TNC, human Lamin A/C, and DAPI. The white lines indicate the location of regenerated tendon-like tissues from human CD29+/CD56+ myogenic progenitors based on human TNC staining. The yellow dashed lines indicate the superimposed location of muscle based on MyHC staining in (**c**). Scale bars, 100 μm. (**c**) Immunofluorescence staining of MyHC and human Lamin A/C. The regenerated tendon-like tissue, the connected muscle, and the surrounding soft tissues were harvested and subjected to continuous cryosection. One of the continuous cryosections was stained for MyHC, which was specifically expressed in skeletal muscle, and human Lamin A/C to label human cells. DAPI indicates nuclei staining. Merge indicates the merged images of MyHC, human Lamin A/C, and DAPI. The white lines indicate the location of regenerated tendon-like tissue from human CD29+/CD56+ myogenic progenitors based on human TNC staining in (**b**). The yellow dashed lines indicate the location of muscle based on MyHC staining. Scale bars, 100 μm. (**d**) Statistical analysis of the percentage of human cells expressing skeletal muscle marker MyHC or tendon marker TNC after being transplanted to the injured tendon. Error bars indicate standard deviation (n=5). (**e**) Immunofluorescence staining of Col I, Col III, and human Lamin A/C. Two months after human cell transplantation, continuous cryosections containing the regenerated tendon-like tissue and native tendon tissue were stained with Col I, Col III, and human Lamin A/C. DAPI indicates the staining of nuclei. T, native tendon tissue; W, wound tendon tissue. Scale bars, 50 μm. (**f**) Immunofluorescence staining of tendon tissue after transplantation of tdTomato+ murine MuSCs. tdTomato indicates the progenies of murine MuSCs. TNC indicates immunofluorescence staining of tendon marker TNC. MyHC indicates immunofluorescence staining of myofiber marker MyHC. DAPI indicates nuclei staining. Merge indicates merged images of tdTomato, TNC, MyHC, and DAPI. The upper panel indicates low-magnification images. Scale bars, 100 μm. The lower panel indicates the amplified images of the region indicated by the white square. Scale bars, 10 μm. (**g**) Statistical analysis of TNC+ tdTomato+ cells in tendon tissue after transplantation of murine MuSCs. Error bars indicate standard deviation (n=5).

The online version of this article includes the following figure supplement(s) for figure 5:

**Figure supplement 1.** Transplantation of human CD29+/CD56+ myogenic progenitors facilitates tendon regeneration, and transplantation of human/murine myogenic progenitors improves muscle regeneration.

# Discussion

Human CD29+/CD56+ myogenic progenitors exhibit dual differentiation potential toward both muscle and tendon lineages. Transplantation of human CD29+/CD56+ myogenic progenitors contributes to injured tendon repair and thus improves locomotor function. Thus, the human CD29+/CD56+ myogenic progenitors could serve as a new source for tendon regeneration.

Tendon disorders widely occur in people of all ages (***Nourissat et al., 2015***). It disrupts the stability and mobility of joint, which deeply affects their locomotor function and quality of life. However, the natural healing of injured tendon is very slow due to hypocellularity and hypovascularity of tendon. The biomechanical property and structural integrity could be hardly completely recovered even with surgical treatment (***Rossi et al., 2019***; ***Bianco et al., 2019***). It is still a great challenge in clinical work to treat tendon injury.

The relative inefficient outcome of routine therapy for tendon injury sparked the exploration of stem cell treatment. Seed cells with the ability to differentiate into tenocytes and secrete paracrine factors to repair tendon injury are preferred. Thus, tendon-derived stem cells, embryonic stem cells, induced pluripotent stem cells, and mesenchymal stem cells have been introduced as seed cells to treat tendon injury (***Lui, 2015***). However, inadequate sources of tendon-derived stem cells, ethical issue and risk of teratoma formation of embryonic stem cells or induced pluripotent stem cell, and heterogeneity of mesenchymal stem cells limit the development of these seed cells for tendon injury treatment. As for muscle progenitors, it is high in proliferation and abundant in sources, and donor site morbidity after muscle harvest was low. Thus, myogenic progenitors might be a promising candidate as seed cells for tendon repair.

The differentiation potential of MuSCs was found to be species-dependent. Human CD29+/CD56+ myogenic progenitors are bipotential adult stem cells. In contrast, murine MuSCs barely have tendon differentiation potential. The species difference might be due to the higher TGFβ signaling level in human CD29+/CD56+ myogenic progenitors. It has been reported that TGFβ/SMAD2/SMAD3 axis plays a crucial role in tendon development and tenogenic differentiation (***Nourissat et al., 2015***; ***Pryce et al., 2009***; ***Berthet et al., 2013***). SMAD2/3-dependent TGFβ signaling also acts as a crucial molecular brake for myogenesis. It could suppress myogenic regulatory factors Myod1 and Myogenin (***Liu et al., 2001***; ***Brennan et al., 1991***), as well as inhibit myotube fusion and muscle regeneration (***Melendez et al., 2021***; ***Girardi et al., 2021***). Thus, the elevated TGFβ signaling may help inhibit the myogenic differentiation ability and stimulate the tenogenic differentiation potential of human CD29+/CD56+ myogenic progenitors under specific microenvironment.

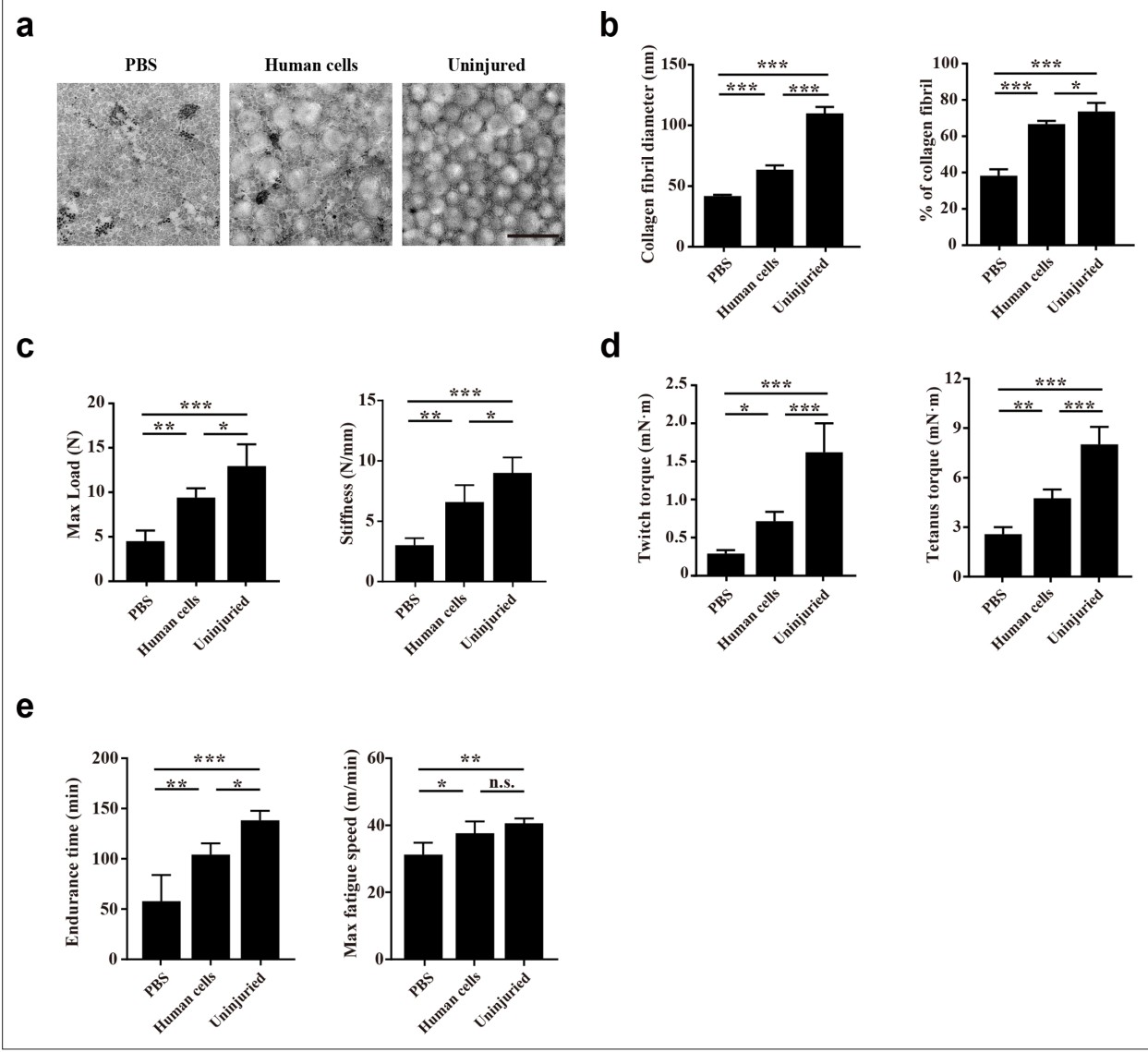

**Figure 6.** Transplantation of human CD29+/CD56+ myogenic progenitors improves the locomotor function after tendon injury. (**a**) TEM images of collagen fibrils in the injured tendon with PBS injection, injured tendon with human CD29+/CD56+ myogenic progenitors transplantation, and uninjured tendon. Scale bars, 500 nm. (**b**) Statistical analysis of the collagen fibril diameter and percentage of collagen fibril area for the injured tendon with PBS injection, injured tendon with human CD29+/CD56+ myogenic progenitors transplantation, and uninjured tendon. Error bars indicate standard deviation (n=5). *p<0.05, ***p<0.001. (**c**) Statistical analysis of the max load and stiffness of the injured tendon with PBS injection, injured tendon with human CD29+/CD56+cmyogenic progenitors transplantation, and uninjured tendon. Error bars indicate standard deviation (n=5). *p<0.05, **p<0.01, ***p<0.001. (**d**) Twitch and tetanus plantarflexion force of the involved limb with PBS injection after tendon injury, human CD29+/CD56+ myogenic progenitors transplantation after tendon injury, and uninjured tendon. Error bars indicate standard deviation (n=5). *p<0.05, **p<0.01, ***p<0.001. (**e**) The results of treadmill exercise for tendon-injured mice with or without human CD29+/CD56+ myogenic progenitors transplantation. The endurance time and max fatigue speed were compared. Error bars indicate standard deviation (n=4). *p<0.05, **p<0.01, ***p<0.001, and n.s. indicates p>0.05.

Studies using mouse models contributed to the majority of our knowledge about MuSCs and laid the foundation for our understanding of mammalian MuSCs. However, humans are dramatically different from mice in many aspects such as size, life span, and manner of motion. It is more demanding for humans to maintain the homeostasis of the locomotion system and the whole organism locomotion ability in a much longer life span and a bigger body size. Though our knowledge about human muscle regeneration is limited, the current studies have revealed multiple differentiation potentials of Pax7+ human myogenic progenitors in skeletal muscles. CD56+CD34+progenitor cells in human skeletal muscle have been reported to have myogenic, osteogenic, and adipogenic activity (***Castiglioni et al.,***

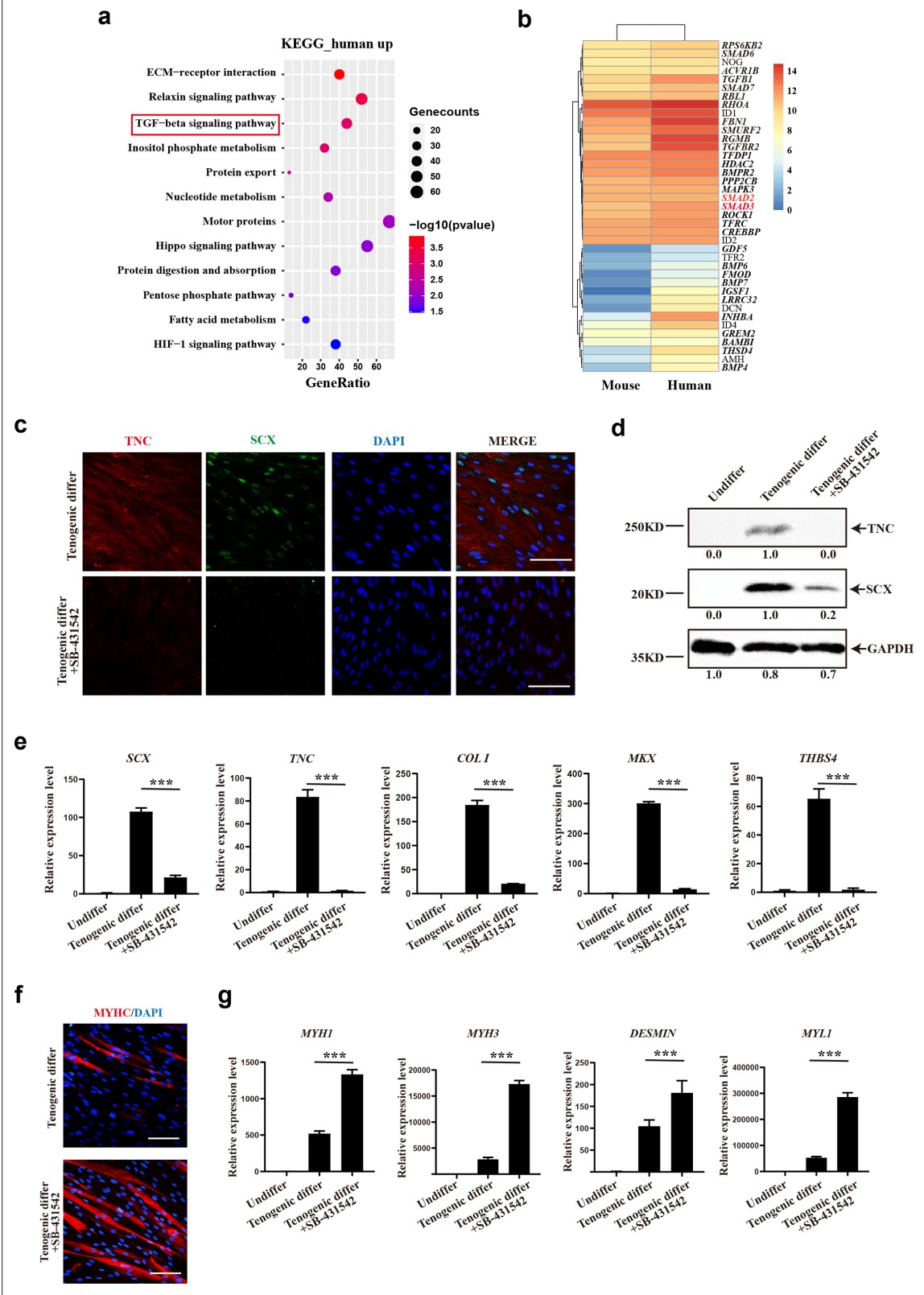

**Figure 7.** TGFβ signaling pathway contributes to tenogenic differentiation of human CD29+/CD56+ myogenic progenitors. (**a**) Bubble chart of KEGG enrichment analysis of upregulated genes in human CD29+/CD56+ myogenic progenitors when compared with mouse muscle stem cells. (**b**) Heatmap of detailed upregulated genes in human CD29+/CD56+ myogenic progenitors which were enriched in TGFβ signaling pathway. (**c**) Immunofluorescence staining of tendon marker TNC and SCX in human CD29+/CD56+ myogenic progenitors induced for tenogenic differentiation with or without TGFβ

*Figure 7 continued on next page*

*Figure 7 continued*

signaling pathway inhibitor SB-431542 for 12 days, respectively. Scale bars, 100 μm. (**d**) Protein levels of TNC and SCX. Human CD29+/CD56+ myogenic progenitors were induced toward tenogenic differentiation with or without TGFβ signaling pathway inhibitor SB-431542 for 12 days, respectively. Total protein was extracted from cells before and after differentiation and subjected to TNC and SCX immunoblotting. GAPDH served as a loading control. (**e**) Relative expression levels of tendon-related genes. RT-qPCR assays were performed with human CD29+/CD56+ myogenic progenitors upon tenogenic differentiation with or without TGFβ signaling pathway inhibitor SB-431542 for 12 days, respectively. GAPDH served as a reference gene. Error bars indicate standard deviation (n=3). \*\*\*p<0.001. (**f**) Immunofluorescence staining of myogenic differentiation marker MyHC in human CD29+/CD56+ myogenic progenitors induced for tenogenic differentiation with or without TGFβ signaling pathway inhibitor SB-431542 for 12 days, respectively. Scale bars, 50 μm. (**g**) Relative expression levels of muscle-related genes. RT-qPCR assays were performed with human CD29+/CD56+ myogenic progenitors upon tenogenic differentiation with or without TGFβ signaling pathway inhibitor SB-431542 for 12 days, respectively. GAPDH served as a reference gene. Error bars indicate standard deviation (n=3). \*\*\*p<0.001.

The online version of this article includes the following source data for figure 7:

**Source data 1.** Labeled raw data for TNC and SCX for human CD29+/CD56+ myogenic progenitors which were induced towards tenogenic differentiation with or without TGFβ signaling pathway inhibitor.

**Source data 2.** Raw data for TNC and SCX for human CD29+/CD56+ myogenic progenitors which were induced towards tenogenic differentiation with or without TGFβ signaling pathway inhibitor.

*2014*; *Pisani et al., 2010a*). The CD56+CD34- progenitor cells in human skeletal muscle have been shown to be free of adipogenic potential (*Pisani et al., 2010b*). Our results suggest that the CD29+/CD56+ myogenic progenitors in skeletal muscles also have tenogenic differentiation ability besides their myogenic differentiation ability. It seems that there are multiple subpopulations of myogenic progenitors in skeletal muscle. They are all capable of muscle regeneration, while with potential to regenerate other components of the motion system such as bone, tendon, and adipocytes. This could be an economic method to maintain the functions of the motion system for the longer life span and more complicated motion manner in human beings.

## Materials and methods
### Animals
Animal care and use were in accordance with the guidelines of the animal facility hosted by Xinhua Hospital affiliated to Shanghai Jiao Tong University School of Medicine, and the operations were approved by the ethical committee of Xinhua Hospital affiliated to the Shanghai Jiao Tong University School of Medicine (approval no. XHEC-F-2021-011). All mice were maintained in specific pathogen-free animal facility in individually ventilated cages with controlled temperature (22±1°C) and light (12 h light/dark cycle). NOD/SCID mice were purchased from the Animal Model Research Center of Nanjing University (cat# SM-019). Pax7CreERT2 and Rosa26-Flox-Stop-Flox-tdTomato mice were purchased from the Jackson Laboratory. All experiments were conducted on 3-month-old adult male mice.

### Human samples
Wasted muscle and remanent tendon were obtained from the patients who underwent knee or shoulder joint surgery procedures. The study was approved by the ethical committee of Xinhua Hospital Affiliated to Shanghai Jiao Tong University, School of Medicine (approval no. XHEC-D-2019-043), and written informed consent was obtained from all donors.

### Isolation of human CD29+/CD56+ myogenic progenitors
Human skeletal muscles were dissected and digested as described previously (*Shao et al., 2023*). Briefly, muscle tissues were cut into small pieces and digested by collagenase II (Worthington Biochemical, 700–800 U/ml, cat# LS004177) for 60 min followed by 30 min of digestion with the mixtures of collagenase II and dispase (Life Technologies, 11U/ml, cat# 17105-041). Digested cells were passed 10 times through a 20-gauge needle. Cell suspension was filtered through a 40 μm cell strainer (BD Falcon, cat# 352340). Erythrocytes were removed by red blood cell lysis buffer (Thermo Fisher Scientific, cat# 00-433-57). The single-cell suspension obtained from human muscle was stained with a cocktail containing PE-Cy5 anti-human CD45 (BD Pharmingen, cat# 555484), Percp-Cy5.5 anti-human CD31 (BioLegend, cat# 303132), AF-488 anti-human CD29 (BioLegend, cat# 303016), and PE

anti-human CD56 (BioLegend, cat# 304606) for 45 min at 4°C. CD45- CD31- CD29+CD56+ myogenic progenitors were sorted by BD Influx sorter (BD Biosciences).

## Isolation of mouse muscle stem cells

For mouse MuSCs isolation, dissected TA muscles were first digested with 10 ml muscle digestion buffer (DMEM containing 1% penicillin/streptomycin, 0.125 mg/ml Dispase II [Roche, 04942078001], and 10 mg/ml Collagenase D [Roche, 11088866001]) for 90 min at 37°C. The digestion was stopped by adding 2 ml of FBS. The digested cells were filtered through 70 µm strainers. Red blood cells were lysed by 7 ml RBC lysis buffer (0.802% NH$_4$Cl, 0.084% NaHCO$_3$, 0.037% EDTA in ddH$_2$O, pH 7.2–7.4) for 30 s, then filter through 40 µm strainers. After staining with antibody cocktails (AF700-anti-mouse Sca-1, PerCP/Cy5.5-anti-mouse CD11b, PerCP/Cy5.5-anti-mouse CD31, PerCP/Cy5.5-anti-mouse CD45, FITC anti-mouse CD34, APC-anti-mouse Integrin α7+), the mononuclear cells were subjected to FACS analysis using Influx (BD Biosciences). The population of PI-CD45-CD11b-CD31-Sca1-CD34+Integrin α7+ cells was collected.

## Primary human tenocytes isolation

Tendon tissues were obtained from the discarded materials of tendon autograft surgery. They were washed with PBS. Epi- and peri-tendon sheath were completely removed. Tenocytes were isolated as described previously (*Bi et al., 2007*). Briefly, the tendons were minced to 1 mm³ pieces and digested with 3 mg/ml collagenase I (Worthington Biochemical, cat# LS004194) in DMEM (Gibco, cat# 11965118) at 37°C for 3 hr with gentle agitation. The digested tissue was filtered through a 40 µm cell strainer (BD Falcon, cat# 352340), and the isolated cells were plated for subsequent analysis.

## Cell culture and differentiation

Primary human CD29+/CD56+ myogenic progenitors were plated in F10 basal medium (Gibco, cat# 11550043) containing 20% FBS (Gibco, cat# 10-013-CV), 2.5 ng/ml bFGF (R&D, cat# 233-FB-025), and 1% Penicillin-Streptomycin (Gibco, cat# 15140-122) on collagen-coated dishes. Mouse MuSCs were plated in F10 basal medium (Gibco, cat# 11550043) containing 20% FBS (Gibco, cat# 10-013-CV), 2.5 ng/ml bFGF (R&D, cat# 233-FB-025), and IL-1α, IL-13, IFN-γ, and TNF-α as described previously (*Fu et al., 2015b*). DMEM (Gibco, cat# 11965118) containing 0.4% Ultroser G (Pall Corporation, cat# 15950-17) and 1% Penicillin-Streptomycin were used to differentiate human muscle stem/progenitor cells (*Ding et al., 2017*). DMEM containing 2% horse serum (HyClone, cat# HYCLSH30074.03HI) and 1% Penicillin-Streptomycin were used to differentiate mouse MuSCs as described previously (*Luo et al., 2015*; *Yin et al., 2018*). DMEM containing 10% FBS, 100 ng/ml GDF5 (R&D, cat# 8340-G5-050), GDF7 (R&D, cat# 8386-G7-050), 0.2 mM ascorbic acid (Sigma-Aldrich, cat# A4403), and 1% Penicillin-Streptomycin were used to induce tendon differentiation (*Otabe et al., 2015*; *Lou et al., 2001*; *Wang et al., 2018*). Primary tenocytes were cultured in DMEM containing 20% FBS, 2.5 ng/ml bFGF (R&D, cat# 233-FB-025), and 1% Penicillin-Streptomycin.

## Single-cell RNA sequencing

The single-cell suspension of mononuclear cells from human skeletal muscles was firstly prepared. PI (Sangon Biotech, cat# E607328) and Hoechst (Sangon Biotech, cat# A601112) were used to sort live cells by FACS. Then the sored cells were washed twice with PBS containing 0.04% BSA, followed by library preparation with Chromium Single Cell 3' Reagent Kits (10X Genomics, cat# 1000121-1000157). The sequencing was performed on Illumina Novaseq 6000 platform (Illumina).

Single-cell RNA-seq data were analyzed using Seurat R (version 3.2.0) package. Cells with less than 200 genes, more than 6000 genes detected, and more than 10% mitochondrial genes were excluded. Three Chinese adult female samples with a total of 57,193 cells were included for subsequent analysis. Sequencing reads for each gene were normalized to total UMIs in each cell to obtain normalized UMI values using 'NormalizeData' function. The 'ScaleData' function was used to scale and center expression levels in the data set for dimensional reduction. Total cell clustering was performed with 'FindClusters' function at a resolution of 0.1, and dimensionality reduction was performed with 'RunUMAP' function (*Qiu et al., 2017*).

## Immunofluorescence staining

Cryosections were fixed in 4% formaldehyde for 15 min, permeabilized in 0.5% Triton X-100 for 15 min, and stained with anti-Pax7 (Developmental Studies Hybridoma Bank), anti-Laminin (Abcam,

cat# ab11575), anti-Lamin A/C (Abcam, cat# ab108595; cat# ab190380), anti-TNC (Abcam, cat# ab108930; cat# ab3970), anti-Scx (Abcam, cat# ab58655), anti-Tnmd (Abcam, cat# ab203676), anti-Col I (Abcam, cat# ab260043), anti-Col III (ABclonal, cat# A0817), or anti-MyHC (Millipore, cat# 05-716) at 4°C overnight, and incubated with Alexa 488-, Alexa 594, or Alexa 647-labeled anti-mouse or anti-rabbit secondary antibodies (Invitrogen, 1:1000) at room temperature for 1 hr. The nuclei were stained with 4,6-diamidino-2-phenylindole (DAPI, Vector Laboratories, cat# H-1200). All images were acquired using Leica SP8 confocal microscope (Leica).

## Gene expression analysis

Total RNA was isolated using TRIzol Reagent (Invitrogen, cat# 15596-018) according to the manufacturer's instruction. GAPDH served as an internal control. The primers for RT-qPCR are listed below:

| Human *GAPDH*-F | 5'-CAAGGCTGAGAACGGGAAGC-3' |
| --- | --- |
| Human *GAPDH*-R | 5'-AGGGGGCAGAGATGATGACC-3' |
| Human *SCX*-F | 5'-AGCGATTCGCAGTTAGGAGG-3' |
| Human *SCX*-R | 5'-GTCTGTACGTCCGTCTGTCC-3' |
| Human *COL I*-F | 5'-GGCTCCTGCTCCTCTTAGCG-3' |
| Human *COL I*-R | 5'-CATGGTACCTGAGGCCGTTC-3' |
| Human *TNC*-F | 5'-GGTGGATGGATTGTGTTCCTGAGA-3' |
| Human *TNC*-R | 5'-CTGTGTCCTTGTCAAAGGTGGAGA-3' |
| Human *THBS4*-F | 5'-TGCTGCCAGTCCTGACAGA-3' |
| Human *THBS4*-R | 5'-GTTTAAGCGTCCCATCACAGTA-3' |
| Human *MKX*-F | 5'-TTCAAGGCAATGCTGAACGG-3' |
| Human *MKX*-R | 5'-CTCCCGCTTTGATGACCGAA-3' |
| Human *COMP*-F | 5'-GATCACGTTCCTGAAAAACACG-3' |
| Human *COMP*-R | 5'-GCTCTCCGTCTGGATGCAG-3' |
| Human *PAX7*-F | 5'-ACCCCTGCCTAACCACATC-3' |
| Human *PAX7*-R | 5'-GCGGCAAAGAATCTTGGAGAC-3' |
| Human *MYF5*-F | 5'-AATTTGGGGACGAGTTTGTG-3' |
| Human *MYF5*-R | 5'-CATGGTGGTGGACTTCCTCT-3' |
| Human *MYOD1*-F | 5'-GACGGCATGATGGACTACAG-3' |
| Human *MYOD1*-R | 5'-AGGCAGTCTAGGCTCGACAC-3' |
| Human *MYOG*-F | 5'-GCTCAGCTCCCTCAACCA-3' |
| Human *MYOG*-R | 5'-GCTGTGAGAGCTGCATTCG-3' |
| Human *MYH1*-F | 5'-GGGAGACCTAAAATTGGCTCAA-3' |
| Human *MYH1*-R | 5'-TTGCAGACCGCTCATTTCAAA-3' |
| Human *MYH3*-F | 5'-ATTGCTTCGTGGTGGACTCAA-3' |
| Human *MYH3*-R | 5'-GGCCATGTCTTCGATCCTGTC-3' |
| Human *DESMIN*-F | 5'-TCGGCTCTAAGGGCTCCTC-3' |
| Human *DESMIN*-R | 5'-CGTGGTCAGAAACTCCTGGTT-3' |
| Human *MYL1*-F | 5'-GTTGAGGGTCTGCGTGTCTTT-3' |
| Human *MYL1*-R | 5'-ACCCAGGGTGGCTAGAACA-3' |
| Mouse *Gapdh*-F | 5'-ACCCAGAAGACTGTGGATGG-3' |
| Mouse *Gapdh*-R | 5'-ACACATTGGGGGTAGGAACA-3' |

*Continued on next page*

*Continued*

| Human *GAPDH*-F | 5'-CAAGGCTGAGAACGGGAAGC-3' |
|---|---|
| Mouse *Col I*-F | 5'-CCAGCGAAGAACTCATACAGC-3' |
| Mouse *Col I*-R | 5'-GGACACCCCTTCTACGTTGT-3' |
| Mouse *Scx*-F | 5'-GAGAACACCCAGCCCAAAC-3' |
| Mouse *Scx*-R | 5'-TCACCCGCCTGTCCATC-3' |
| Mouse *Tnc*-F | 5'-GAGCCCCTTTGCCTCAACAA-3' |
| Mouse *Tnc*-R | 5'-CTTCGCCCGTGAAACCTTCTT-3' |
| Mouse *Mkx*-F | 5'-TGGTTTCCTGGACAATCCACA-3' |
| Mouse *Mkx*-R | 5'-CGCTTATGCCTTACCTTCCCTC-3' |
| Mouse *Thbs4*-F | 5'-GCCACAAGCACAGGAGACTTT-3' |
| Mouse *Thbs4*-R | 5'-TGACCTGCTGCCTCAGAAGA-3' |
| Mouse *MyoG*-F | GAGACATCCCCCTATTTCTACCA |
| Mouse *MyoG*-R | GCTCAGTCCGCTCATAGCC |
| Mouse *Myh1*-F | 5'-GCGAATCGAGGCTCAGAACAA-3' |
| Mouse *Myh1*-R | 5'-GTAGTTCCGCCTTCGGTCTTG-3' |
| Mouse *Myh3*-F | 5'-ATGAGTAGCGACACCGAGATG-3' |
| Mouse *Myh3*-R | 5'-ACAAAGCAGTAGGTTTTGGCAT-3' |

## Cloning assay

Primary human CD29+/CD56+ myogenic progenitors were first sorted in 96-well plates with density of single cell per well. Tenogenic induction or myogenic induction was performed after proliferation for each well. After induction, the immunofluorescence staining of SCX or MyHC was performed in each well. The differentiation efficiency was determined by calculating the ratio of total wells with positive fluorescence signal to total wells with alive cells.

## RNA-sequencing

Total RNA was isolated using TRIzol Reagent (Invitrogen, cat# 15596-018) according to the manufacturer's instruction. mRNA was enriched with magnetic oligo (dT) beads (New England Biolabs, cat# S1419S). The cDNA library was constructed with mean inserts of 200 bp with non-stranded library preparation using NEBNext Ultra RNA Library Prep Kit for Illumina (New England Biolabs, cat# E7530L). Sequencing was performed by a paired-end 125 cycles rapid run on the Illumina HiSeq2500. Sequencing data were filtered by SolexaAQ (Q>20 and length ≥25 bp) (*Cox et al., 2010*). The adapter sequences and low-quality segments (Phred Quality Score<20) were trimmed using Cutadapt. Paired-end clean reads were then mapped to the reference genome GRCh38.98 using HISAT2. Htseq-count was used to quantify the gene expression value (*Anders et al., 2015*). Read count of each gene was normalized using FPKM (fragments per kilo bases per million fragments). Differential expression (DE) analysis was performed using DESeq, and significant DE genes were defined as those with absolute log2FoldChange >1 and p<0.05. Heatmap and volcano plot of DE genes were generated using the R package Pheatmap and EnhancedVolcano, respectively. Gene enrichment analysis was conducted using R package topGO, with p<0.05 as the cut-off. The differential expression analysis for RNA-seq data between human CD29+/CD56+ myogenic progenitors and mouse MuSCs was performed according to the literature (*Zhou et al., 2019*).

## Cell transplantation in TA muscle

A single dose of 18 Grey irradiation was administered to the hind legs of the recipient NOD/SCID mice. TA muscle was injured by injecting 15 µl of 10 µM CTX (Sigma), and 50,000 human CD29+/CD56+ myogenic progenitors or 50,000 murine MuSCs suspended in 10 µl PBS were injected intramuscularly to the injury site as described (*Ding et al., 2017*).

## Tendon injury for lineage tracing

Pax7$^{CreERT2}$:Rosa26$^{tdTomato}$ mice were obtained and MuSC was labeled with tdTomato as aforementioned. A mouse surgical model was established using dedicated instruments to replicate procedures analogous to previously reported human tendon surgeries (*Shao et al., 2020*). Firstly, distal gastrocnemius tendon was woven with a 6-0 polypropylene non-absorbable suture (PROLENE, cat# EH7242H) and released by microsurgical scissor. Then, a dedicated mini tendon stripper was introduced over the free end of distal medial gastrocnemius tendon. The medial gastrocnemius tendon could be totally removed after advancement of tendon stripper. To further activate MuSCs to guarantee that sufficient amount of activated MuSCs were available around the tendon injury site, 15 µl of 10 µM CTX CTX was injected at the muscle adjacent to the sites where the tendon was removed to induce more muscle injuries and further activate MuSCs. Then the subcutaneous tissue and skin were sutured.

## Tendon injury and cell transplantation

An approximately 1.5 mm long and 0.5 mm wide transverse incision was performed at 5 mm from the calcaneus in Achilles tendon for recipient NOD/SCID mice. 50,000 human CD29+/CD56+ myogenic progenitors or mouse MuSCs resuspended in 20 µl hydrogel were injected to the injury site with 28-gauge needles. PBS mixed with 20 µl hydrogel were injected as control.

## Biomechanical analysis

Two months after tendon injury, the tendons were harvested to biomechanical analysis using a universal tester (Instron 3345 load system, USA). The grippers were gradually moved apart with the speed of 5 mm/min until the tested tendon was completely ruptured. Then the max load was obtained and documented. The slope of the stress–strain curve was defined as stiffness. All these data were automatically presented in Instron 3345 load system.

## TEM examination

After washing the selected target tendon tissues with PBS, tissue fixation was performed for more than 2 hr. Post-fixation was conducted by 1% $OsO_4$ (Ted Pella Inc) in 0.1 M PB for 1–2 hr. Then dehydration and drying were performed, and target tendon samples were subsequently attached to metallic stubs for conductive metal coating. Images were obtained by TEM (HITACHI, SU8100), and at least 1000 collagen fibrils were evaluated for each sample. Density of fibrils was evaluated by the percentage of collagen fibril-containing area.

## Treadmill test

Mouse treadmill test was performed in Exer-3/6 treadmill apparatus with electrical stimulus. Treadmill exercise began with a 5 min warm-up, then each mouse was evaluated with specific protocol. For endurance test, mice ran on the treadmill at the constant speed of 22 m/min until exhaustion. For exhaustion test, the treadmill speed was increased (2 m/min each 3 min) at the beginning speed at 18 m/min until exhaustion. Exhaustion was defined as inability to run on the treadmill for longer than10 seconds.

## In vivo muscle force analysis

The 1300A 3-in-1 whole animal system (Aurora Scientific) was used for in vivo muscle force analysis. Mice were first anesthetized and kept warm by a heat lamp. The foot was placed on a footplate and kept perpendicular to the tibia. The peroneal branch of the sciatic nerve was stimulated to evaluate the plantarflexion strength. Five repetitive tests were performed for each limb, and DMA software (Aurora Scientific) was used for results analysis.

## Statistical analysis

At least three biological replicates and technical repeats were performed in each experimental group. All experiments were analyzed and evaluated by investigators in a blinded manner. Error bars indicate standard deviation. Two-tailed unpaired Student's t-tests were used when variances were similar (tested with F-test) for comparison of two independent groups. One-way ANOVA followed by Dunnett's post-test or Tukey's post-test were used for multiple comparisons. Shapiro–Wilk tests were performed to determine data normality. Statistical analysis was performed in GraphPad Prism 7

(GraphPad Software, San Diego, USA) or SPSS version 19.0 for Windows (SPSS Inc, Chicago, IL, USA). p value <0.05 was considered significant. Data were presented as mean ± standard unless stated otherwise.

## Acknowledgements

We thank Dr. Dangsheng Li for helpful discussions, the National Protein Science Center (Shanghai) for help with FACS sorting, and the cell biology facility of SIBCB for help with imaging and FACS analysis. This work was supported by the Strategic Priority Research Program of the Chinese Academy of Science (XDA16020400) and the National Science Foundation of China (32170804, 81871096, 82372384, 82302657).

## Additional information

### Funding

| Funder | Grant reference number | Author |
|---|---|---|
| National Natural Science Foundation of China | 32170804 | Ping Hu |
| National Natural Science Foundation of China | 81871096 | Ping Hu |
| National Natural Science Foundation of China | 82372384 | Jianhua Wang |
| National Natural Science Foundation of China | 82302657 | Xiexiang Shao |
| Strategic Priority Research Program of the Chinese Academy of Science | XDA16020400 | Ping Hu |

The funders had no role in study design, data collection and interpretation, or the decision to submit the work for publication.

### Author contributions

Xiexiang Shao, Conceptualization, Resources, Data curation, Software, Formal analysis, Supervision, Validation, Investigation, Visualization, Methodology, Writing – original draft, Project administration, Writing – review and editing; Xingzuan Lin, Conceptualization, Data curation, Software, Formal analysis, Validation, Investigation, Visualization, Methodology, Writing – original draft, Writing – review and editing; Hao Zhou, Data curation, Formal analysis, Investigation, Writing – original draft, Writing – review and editing; Minghui Wang, Formal analysis, Writing – original draft; Lili Han, Jianhua Wang, Data curation, Writing – original draft, Writing – review and editing; Xin Fu, Investigation, Writing – review and editing; Sheng Li, Shenao Zhou, Formal analysis, Writing – review and editing; Siyuan Zhu, Data curation, Formal analysis, Visualization, Writing – original draft, Writing – review and editing; Wenjun Yang, Zhanghua Li, Funding acquisition, Writing – original draft, Writing – review and editing; Ping Hu, Funding acquisition, Methodology, Writing – original draft, Writing – review and editing

### Author ORCIDs

Xiexiang Shao (ID) https://orcid.org/0000-0003-4837-1305
Xingzuan Lin (ID) http://orcid.org/0009-0006-6063-666X
Jianhua Wang (ID) https://orcid.org/0000-0001-6831-9593

### Ethics

The study was approved by the ethical committee of Xinhua Hospital Affiliated to Shanghai Jiao Tong University, School of Medicine (Approval No. XHEC-D-2019-043) and written informed consents were obtained from all donors.

The protocol was approved by the Committee on the Ethics of Animal Experiments of Xinhua Hospital affiliated to Shanghai Jiao Tong University School of Medicine (Approval No. XHEC-F-2021-011).

Reviewer #3 (Public review): https://doi.org/10.7554/eLife.98636.3.sa1
Author response https://doi.org/10.7554/eLife.98636.3.sa2

## Additional files

### Supplementary files
MDAR checklist

### Data availability
The complete sequencing data have been uploaded on to Sequence Read Archive database (PRJNA1178160, PRJNA1012476 and PRJNA1012828). The raw data supporting the findings of this study have been deposited in Dryad under the following DOI: https://doi.org/10.5061/dryad. b2rbnzss8.

The following datasets were generated:

| Author(s) | Year | Dataset title | Dataset URL | Database and Identifier |
|---|---|---|---|---|
| Wang J | 2024 | scRNA-seq of human muscle cells | https://www.ncbi.nlm. nih.gov/bioproject/? term=PRJNA1178160 | NCBI BioProject, PRJNA1178160 |
| Wang J | 2023 | RNA-seq of human muscle stem cells | https://www.ncbi.nlm. nih.gov/bioproject/? term=PRJNA1012476 | NCBI BioProject, PRJNA1012476 |
| Wang J | 2023 | RNA-seq of mouse muscle stem cells | https://www.ncbi.nlm. nih.gov/bioproject/? term=PRJNA1012828 | NCBI BioProject, PRJNA1012828 |
| Shao X, Lin X, Zhou H, Wang M, Han L, Fu X, Li S, Zhu S, Zhou S, Yang W, Wang J, Li Z, Hu P | 2025 | Human CD29+/CD56+ myogenic progenitors display tenogenic differentiation potential and facilitate tendon regeneration | https://doi.org/10. 5061/dryad.b2rbnzss8 | Dryad, 10.5061/dryad. b2rbnzss8 |

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
