## [Editor Report · eLife Assessment]

The authors demonstrate the **valuable** discovery that human CD29+/CD56+ myogenic progenitors can differentiate into tendon through the TGFβ pathway, addressing mouse and human interspecies differences in regard to the potential of muscle stem cells. The in vivo transplantation experiments provide **convincing** evidence for the conclusion as human CD29+/CD56+ myogenic progenitors contribute to tendon regeneration, resulting in functional recovery in mouse model. The authors’ approach can be used for the development of cell therapy for tendon-injured patients.

---

## [Referee Report · Reviewer #3 (Public review)]

Summary:

The authors have thoroughly addressed all my concerns. The revised version of the current manuscript is solid now. It's very interesting that there is bi-potential ability of human CD29/CD56+ myogenic progenitors. The current study substantiates the medical translational potential for human CD29/CD56+ myogenic progenitors in promoting tendon regeneration.

Strengths:

CD29+/CD56+ stem/progenitor cells were transplanted into immunodeficient mice with a tendon injury, and human cells expressing tenogenic markers contributed to the repair of the injured tendon. Furthermore, the authors also show better tendon biomechanical properties and plantarflexion force after transplantation.

Weaknesses:

None. The authors have thoroughly addressed all my concerns.

---

## [Author Response]

The following is the authors’ response to the previous reviews.’

**Public Reviews:**

**Reviewer #1 (Public Review):**
For the colony analysis, it is unclear from the methods and main text whether the initial individual sorted colonies were split and subject to different conditions to support the claim of bi-potency. The finding that 40% of colonies displayed tenogenic differentiation, may instead suggest heterogeneity of the sorted progenitor population. The methods as currently described, suggest that two different plates were subject to different induction conditions. It is therefore difficult to assess the strength of the claim of bi-potency.

Thanks for your valuable comment. We are sorry for the confusing illustration of colony assay. In fact, we first obtained CD29+/CD56+ myogenic progenitors by FACs. Then these freshly isolated cells were randomly seeded to 96-well plate with density of 1 cell/well. Subsequently, the single cell in each plate was cultured with growth medium to form colonies for ten days. Then myogenic induction was performed in three 96-well plates and tenogenic induction was performed in another three 96-well plates for subsequent analyses. We agree with your point that the sorted cell population could be heterogeneous myogenic progenitors. The result showed over 95% colonies successfully differentiated into myotubes, while 40% of colonies displayed tenogenic differentiation (Fig. 2g). Since the freshly obtained CD29+/CD56+ myogenic progenitors were randomly seeded for tenogenic induction or myogenic induction, the undifferentiated cells in each group were considered as the same sample. Furthermore, the optimal tenogenic differentiation condition for these cells was still waiting for investigation. Thus, we believe the colony analysis combined with the data in Figure 1 and Figure 2 could indicate the bi-potency for human CD29+/CD56+ myogenic progenitors.

This group uses the well-established CD56+/CD29+ sorting strategy to isolate muscle progenitor cells, however recent work has identified transcriptional heterogeneity within these human satellite cells (ie Barruet et al, eLife 2020). Given that they identify a tenocyte population in their human muscle biopsy in Figure 1a, it is critical to understand the heterogeneity contained within the population of human progenitors captured by the authors' FACS strategy and whether tenocytes contained within the muscle biopsy are also CD56+/CD29+.

Thanks for your constructive suggestion. We have included more samples to perform scRNA-seq and reanalyzed the data. The scRNA-seq data revealed that all the CD29+/CD56+ cells were myogenic progenitors, which occupied 19.3% of all the myogenic progenitors (Fig. 1e). However, there existed no tenocytes with CD29+/CD56+ (Fig. 1d), and tenocytes made up only a small percentage (0.06%) of all the mononuclear cells. Thus, human CD29+/CD56+ cells are myogenic progenitors, and tenocytes contained within the muscle biopsy are not CD56+/CD29+. In addition, both published research and our results indicated the heterogeneity of CD29+/CD56+ myogenic progenitors. Since the main purpose of current study was to investigate the tenogenic differentiation potential of CD29+/CD56+ myogenic progenitors, the heterogeneity in CD29+/CD56+ myogenic progenitors should be investigated in the further study.

The bulk RNA sequencing data presented in Figure 3 to contrast the expression of progenitor cells under different differentiation conditions are not sufficiently convincing. In particular, it is unclear whether more than one sample was used for the RNAseq analyses shown in Figure 3. The volcano plots have many genes aligned on distinct curves suggesting that there are few replicates or low expression. There is also a concern that the sorted cells may contain tenocytes as tendon genes SCX, MKX, and THBS4 were among the genes upregulated in the myogenic differentiation conditions (shown in Figure 3b).

Thanks for your comment. Each group consisted of three samples for RNAseq analyses. We are sorry there existed a minor analysis mistake in Fig. 3b and Fig. 3c, which have been reanalyzed in the revised version. There was no significantly difference of tendon related marker genes after myogenic differentiation (Fig. 3b), while these tenogenic genes were significantly up-regulated after tenogenic induction (Fig. 3c). As for contamination of tenocytes, scRNA-seq data showed there were no tenocytes with both CD29 and CD56 positive (please see response to Comment 2). And almost all the obtained cells highly expressed myogenic progenitors markers PAX7/MYOD1/MYF5 (Fig. 1f-g). Low expression levels of tendon markers were identified in these cells (Fig. 2a-c). Furthermore, although tendon genes slightly upregulated in myogenic differentiation conditions, these markers dramatically upregulated in tenogenic differentiation conditions (Fig. 2c). Thus, we believe the bulk RNA sequencing data could add the evidence of tenogenic differentiation ability of human CD29+/CD56+ myogenic progenitors.

**Reviewer #2 (Public Review):**
scRNAseq assay using total mononuclear cell population did not provide meaningful insight that enriched knowledge on CD56+/CD29+ cell population. CD56+/CD29+ cells information may have been lost due to the minority identity of these cells in the total skeletal muscle mononuclear population, especially given the total cell number used for scRNAseq was very low and no information on participant number and repeat sample number used for this assay. Using this data to claim a stem cell lineage relationship for MuSCs and tenocytes may not convincing, as seeing both cell types in the total muscle mononuclear population does not establish a lineage connection between them.

Thanks for your constructive suggestion. We have included more samples to perform scRNA-seq and reanalyzed the data. Three samples with a total of 57,193 cells were included for analysis. As you can see in Fig. 1d and 1e, the joint expression analysis revealed that all the CD29+/CD56+ cells were myogenic progenitors, which occupied 19.3% of all the myogenic progenitors. In addition, we agree with your comment that the pseudotime analysis could be a bit misleading as the nature of computational biology with pseudotime plots, so we deleted this assay.

The TGF-b pathway assay uses a small molecular inhibitor of TGF-b to probe Smad2/3. The assay conclusion regarding Smad2/3 pathway responsible for tenocyte differentiation may be overinterpretation without Smad2/3 specific inhibitors being applied in the experiments.

Thanks for your comment. We agree with your comment and we have revised it in the revision version (Figure 7, Line 306-326).

**Reviewer #3 (Public Review):**
This dual differentiation capability was not observed in mouse muscle stem cells.

Thanks for your comment. We have explored the tenogenic differentiation potential of mouse MuSCs both in vivo and in vitro. However, low tenogenic differentiation ability was revealed (Figure 4), which might be due to species diversity. Maybe it is more demanding for humans to maintain the homeostasis of the locomotion system and the whole organism locomotion ability in much longer life span and bigger body size. Thus, the current study also indicated that anima studies may not clinically relevant when investigating human diseases.

**Recommendations For The Authors:**

**Reviewer #1 (Recommendations For The Authors):**
The methods section contained insufficient details for sample tissue for many methods, including the single cell analysis, RNA FISH, and for in vivo cardiotoxin treatment. ie. how were the samples subclustered for the monocle pseudotime analysis; how many cells were counted in the FISH shown in Fig 1e/f, does the n=5 refer to tissue sections or biological replicates?; for the double injury, what was the cardiotoxin dose?

Thanks for your comment. Three samples and a total 57,193 cells were analyzed in single cell analysis (Line 464). We deleted RNA FISH assay data because it provided limited information to prove bipotential ability of human CD29+/CD56+ myogenic progenitors. In addition, since the pseudotime analysis could be a bit misleading as the nature of computational biology with pseudotime plots, we also deleted this assay. For the double injury, 15μl of 10μM cardiotoxin was used for lineage tracing (Line 533).

Additionally, the RNA sequencing datasets are not currently publicly available under the accession numbers provided.

The raw data of RNA sequencing has been uploaded in NCBI (accession number: PRJNA1178160, PRJNA1012476 and PRJNA1012828), and these data will be released immediately after publication.

The poor resolution of 1d makes it impossible to read any of the gene names or interpret the expression profiles of their proposed trajectories.

Since the pseudotime analysis could be a bit misleading as the nature of computational biology with pseudotime plots, we deleted this assay.

What does the color key for 3a refer to? It is not indicated in the figure or legend.

Thanks for your comment. The color key for 3a refer to “Scaled expression values”, which has been added in the revised version.

scRNAseq of the sorted CD29/56+ population could help uncover possible cell heterogeneity within these muscle progenitors and which sub-populations of myogenic progenitor cells have tenogenic potential.

Thanks for your valuable suggestion. We included more cells from three biological repetitions to perform scRNA-seq and found that CD29/CD56+ cells were absolutely from myogenic progenitors (Fig. 1d and 1e). We agree with you that additional scRNAseq will be helpful to clarify the possible cell heterogeneity within these muscle progenitors. Since the main scope of current study is to investigate the biopotential of CD29/CD56+ myogenic progenitors, analysis of scRNAseq of the sorted CD29/56+ population would be performed in the further study for further exploration.

Typos: Line 459 sored cells... preparasion with Chromium Single Cell 3' Reagent Kits (10X genomics, cat# 1000121-1000157). Figure 4E - typo in the word tamoxifen.

Thanks for your valuable suggestion. We are sorry for the typos and have revised these typos (Line 459 and Fig. 4e).

**Reviewer #2 (Recommendations For The Authors):**
(1) scRNAseq is performed in total mononuclear cells isolated from human skeletal muscle. The cell number (around 15000 cells) seems very low for this assay, given the CD56+/CD29+ cells are a minority population in this sequencing, the data does not seem to provide meaningful insight into the MuSC cell identities. No information on sample numbers and number of patient participants can be found in the paper.

Thanks for your comment. We added more cells to reanalyze the data in the revised manuscript. Three samples with a total of 57,193 cells were analyzed (Line 464). The joint expression analysis revealed that all the CD29+/CD56+ cells were myogenic progenitors, which occupied 19.3% of all the myogenic progenitors (Fig. 1d and 1e). These scRNA-seq data combined with functional experiment confirmed the MuSC cell identity of CD29+/CD56+ cells from mononuclear cells.

In this regard, the paragraph starts with "To confirm the single cell analysis results, we first isolated myogenic progenitor cells from human muscle biopsy using FACS as described previously" which is misleading as the seRNAseq is not the result of the sorted cells. Please reword this paragraph to clarify.

The related paragraph has been reworded (Line 84-95).

Similarly, the existence of myocytes and tenocytes in scRNAseq does not necessarily prove a stem cell and mature cell lineage relationship. Please edit the wording to avoid overinterpretation.

Thanks for your reminding. Since the pseudotime analysis could be a bit misleading as the nature of computational biology with pseudotime plots, we deleted this assay.

(2) The in vitro differentiation assays are well performed, which included bulk culture and clonal culture. The efficiencies of those two assays seem to have discrepancies which may need clarification. Again, no sample numbers and repeats have been informed.

Since the tendon differentiation period for bulk culture was 12 days, those myotubes fused by CD29+/CD56+ myogenic progenitors with only myogenic differentiation potential will be no longer alive. Thus, the efficiency of bulk culture seemed higher than that in clonal culture. As stated in statistical analysis, at least three biological replicates and technical repeats were performed in each experimental group (Line 577).

In these paragraphs, terminologies including MuSCs, myogenic progenitors, CD56+/CD29+, and Pax7+ are interchangeably used, which generates confusion while reading. It is probably best to consistently use the cell sorting markers markers to address this cell population, throughout the paper.

Thanks for your constructive suggestion. The cell population was consistently named as CD29+/CD56+ myogenic progenitors throughout the paper.

Information on the proliferation rate and expansion of the MuSCs would be useful but not provided.

Thanks for your comment. The analysis of cell proliferation was added in Figure 1 (Fig. 1h).

The murine cell differentiation assays are not as convincing as the human study. The assay regarding "mouse muscle CD29+/CD56+ cells were isolated for tenogenic induction. However, very few mouse muscle CD29+/CD56+ cells expressed myogenic progenitor cell marker Pax7, MyoD1 and Vcam1" does not add any value to the work as those markers are not mouse MuSC markers to start with.

Thanks for your comment. The experiments concerning mouse muscle CD29+/CD56+ cells have been deleted to avoid misleading.

The Pax7-cre-TdTomato assay was also not convincing, as a negative finding may not be the best proof of absence.

Thanks for your comment. Pax7 positive cells could consistently express TdTomato for lineage tracing. In current study, large amount of tdTomato+ myofibers were observed after muscle injury (SFig. 2c-d), suggesting that the tracing system works well. However, less than 0.2% tendon cells originated from TdTomato+ MuSCs were observed even four months after tendon removal (Fig. 4f-g). When comparing in vivo data between murine MuSCs and human CD29+/CD56+ myogenic progenitors, we believe these data could indicate the poor tendon differentiation abilities of murine MuSCs.

(5) TGFb as a pathway of smad2/3 mediated tenocyte differentiation assays were well done albeit not novel. Using TGFb universal inhibitor may not accurately state the pathways were due to SMAD2/3 inhibition either.

We agree with your comment and the conclusion concerning SMAD2/3 has been deleted throughout the manuscript.

The paper also needs thorough proofreading. Currently, typographic, grammatical, and logical sequences of writing do not lend the paper to easy reading.(1) Figure 1K and 1I have similar legends but presumably K is referring to MuSC and I is referring to differentiated cells.(2) Tenogenic and myogenic induction should be changed to tenogenic/myogenic differentiation as they are the cells at the end of differentiation.(3) Figure 6, it is not clear how the "human cells" are calculated in this assay.

Thanks for your constructive comment. (1) The figure legends in Figure1 have been revised (Line 797-804). (2) Tenogenic and myogenic induction have been changed to tenogenic/myogenic differentiation manuscript when they are referring to cells at the end of differentiation (Fig.1, Fig.2, Fig.3, Fig.4, Fig.7 and SFig.1). (3) In Figure 6, “human cells” is referring to those injured tendons with transplantation of human CD29+/CD56+ myogenic progenitors. To evaluate the function of human CD29+/CD56+ myogenic progenitors, PBS group was set as negative control and uninjured group was set as normal control.

**Reviewer #3 (Recommendations For The Authors):**
(1) The full extent of the differentiation potential of CD29+/CD56+ stem/progenitor cells has not been thoroughly evaluated. There can also exist heterotopic ossification in injured tendon sites. Thus, it remains unclear whether these cells are truly bipotent as the authors claim, or can they differentiate into chondrocytes and osteoblasts.

Thanks for your comment. The current study focused on the tenogenic differentiation potential of CD29+/CD56+ myogenic progenitors, so the research priority was the bipotential ability of CD29+/CD56+ myogenic progenitors. We agree with you that chondrogenic and osteogenic ability of CD29+/CD56+ myogenic progenitors is also important and would investigate it in the further study.

(2) In Figure 3, the GO analysis also shows increased enrichment of muscle-related terms including muscle contraction and filament. Please clarify it.

The tenogenic differentiation efficiency of CD29+/CD56+ myogenic progenitors was about 40% in clonal assay. Some cells would myogenically differentiated under this tenogenic induction system. Thus, the GO analysis could also enrich muscle related terms including muscle contraction and filament.

(3) The authors use TNC staining to evaluate cell transplantation. My concern is whether the TNC expression is specific to the tendon site, or do engrafted human cells also express TNC in other sites such as muscle?

TNC is one of a well-known tendon-related markers. As you can see in Figure 6b and Figure 6c, although some human cells (labeled by Lamin A/C) were engrafted in muscle tissue area (labeled by MyHC), these engrafted human cells didn’t express TNC in muscle. In addition, we also used tendon related markers SCX and TNMD to confirm the tenogenic differentiation ability of engrafted human cells in vivo (SFig. 3a and 3b).

(4) The authors demonstrate that CD29+/CD56+ human stem/progenitor cells could efficiently transplant and contribute to myofiber regeneration in vivo. However, why were only a few transplanted human cells differentiating into myofiber (labeled by MyHC) in the tenon injury model even with CTX injection?

Thanks for your comment. Since skeletal muscle is able to regenerate with in situ muscle progenitor cells, regeneration of injured muscle by CTX injection was dependent on not only CD29+/CD56+ myogenic progenitors, but also native murine MuSCs. Thus, it is reasonable that there were only a few transplanted human cells differentiating into myofiber (labeled by MyHC) in the tenon injury model even with CTX injection.

(5) Figure 7 shows the crucial role of TGFB/SMAD signaling for the tenogenesis of human CD29+/CD56+ stem/progenitor cells. However, can TGFB/SMAD signaling activation facilitate the tenogenic differentiation of mouse MuSCs? This point is crucial to clarify the difference of MuSCs between different species.

Thanks for your valuable suggestion. We did a series of pilot assays to investigate the effect of TGFβ signaling activation to facilitate tenogenic differentiation of mouse MuSCs (Author response image 1). As you can see, activating TGFβ by SRI-011381 could slightly increase the expression of tenogenic markers of murine MuSCs. It’s an interesting topic and we would investigate it in the further study.

**Author response image 1. sa2fig1:** TGFβ signaling pathway slightly elevated tenogenic differentiation ability of murine MuSCs. (a) Immunofluorescence staining of tendon marker Scx and Tnc in murine MuSCs induced for tenogenic differentiation with or without TGFβ signaling pathway agonist SRI-011381, respectively. Scale bars, 50 µm. (b) Quantification of Scx and Tnc fluorescent intensity in murine MuSCs undergone tenogenic induction with or without TGFβ signaling pathway agonist SRI-011381, respectively. Error bars indicated standard deviation (n=5). (c) Protein levels of Tnc and Scx. Murine MuSCs were induced towards tenogenic differentiation with or without TGFβ signaling pathway agonist SRI-011381. Total protein was extracted from cells before and after differentiation and subjected for Tnc and Scx immunoblotting. GAPDH was served as loading control.

(6) Please quantify the WB blot data throughout the manuscript.

Thanks for your comment. The WB blot data has been quantified throughout the manuscript.

(7) The data of RT-qPCR should indicate what the fold changes in relative to throughout the manuscript.

Thanks for your comment. The sentence “GAPDH was served as reference gene” was added in the figure legends to illustrate RT-qPCR results.